

# Two portals to GeV sterile neutrinos: Dipole versus mixing

Enrico Bertuzzo[1,2] and Michele Frigerio[3]

**1** Dipartimento di Scienze Fisiche, Informatiche e Matematiche,
Università degli Studi di Modena e Reggio Emilia,
Via Campi 213/A, I-41125 Modena, Italy
**2** INFN sezione di Bologna, via Irnerio 46, 40126 Bologna, Italy
**3** Laboratoire Charles Coulomb (L2C), University of Montpellier,
CNRS, Montpellier, France

## Abstract

Massive sterile neutrinos, also known as heavy neutral leptons, can have a mixing with active neutrinos, $\theta$, as well as a dipole coupling to the photon, $d$. We study the interplay between these two portals, considering the production from meson decays of sterile neutrinos with mass $0.1$ GeV $\lesssim M_N \lesssim 10$ GeV, at beam-dump facilities such as NA62 and SHiP, and at the FASER2 experiment. These sterile neutrinos can be long-lived and decay into a photon in a distant detector, via the dipole operator. We find that all these experiments will be sensitive to values of $d$ which are presently unconstrained. The experimental reach varies strongly with the mass $M_N$ and the mixing $\theta$, and one observes specific correlations with the flavour of active neutrinos. The SHiP experiment will mark a jump in sensitivity: (i) it will probe a sterile dipole as small as $d \sim 10^{-8}$ GeV$^{-1}$, thus testing new physics well above the electroweak scale; (ii) it may detect the active-sterile dipole to the level predicted by electroweak loops, if $\theta$ is close to the present bound.

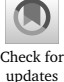

# 1  Introduction

Among possible fermionic extensions of the Standard Model (SM), a special role is played by gauge-singlet fermions, also known as sterile neutrinos $N$. Their mass scale $M_N$ is essentially unconstrained: they can be as heavy as the quantum gravity scale $M_{Planck} \sim 10^{19}$ GeV, as light as the electroweak scale $M_W \sim 10^2$ GeV, or even massless, as long as their couplings to the SM are sufficiently suppressed. They may have Yukawa couplings to the SM operator $HL$, made of a Higgs and a lepton doublet: in this case they mix with active neutrinos and they contribute to their mass. Such contribution can be the dominant one in the window $m_\nu \sim 0.1$ eV $\lesssim M_N \lesssim 10^{15}$ GeV. When sterile neutrinos lie within the reach of collider experiments, say for 0.1 GeV $\lesssim M_N \lesssim 10^4$ GeV, they are often dubbed heavy neutral leptons (HNLs).

In this mass range, experimental limits on the mixing with active neutrinos are particularly strong (see e.g. the reviews [1, 2] and the references therein) and higher-dimensional interactions may be phenomenologically relevant.[1] In particular, even though sterile neutrinos are singlets with respect to the SM gauge group, they are allowed to couple to gauge bosons via quantum corrections, but also through effective higher-dimensional operators. The most relevant such operator is a dimension-five dipole coupling between $N$ and the hypercharge field strength $B_{\mu\nu}$, which would be generated in any ultraviolet (UV) theory where $N$ couples to heavy states carrying hypercharge. The sterile neutrino dipole operator was first introduced in [4, 5]. This interaction may literally shed light on sterile neutrinos, as they could be produced and/or detected by their dipole coupling to the photon.

Since the phenomenology of GeV-scale sterile neutrinos is determined by two small parameters, the mixing and the dipole, they are expected to have a small decay width. Therefore, experiments aiming at detecting long-lived particles may be particularly effective in probing the parameter space, since they are designed to detect light, weakly-coupled new physics. These experiments will be the main focus of our work. More in detail, we consider $N$ production from meson decays, which in turn are produced either in beam-dump experiments or in proton-proton collisions [6–10], and we focus on the detection of the photon emitted in the subsequent $N$ decays, in a detector placed at a large distance from the interaction point.

The relevant mass region is 0.1 GeV $\lesssim M_N \lesssim 10$ GeV: we will see that lighter sterile neutrinos are subject to other constraints and, moreover, the sensitivity to the dipole tends to decrease with the mass; on the other hand, the upper bound on the sterile mass comes from

---

[1]A similar reasoning has been used in the case of the dark photon in [3], where it was shown that the effect of higher-dimensional operators can compete with a small kinetic mixing, and drastically change the phenomenology.

the heaviest available mesons. For sterile neutrinos without a dipole, there is a vast literature setting bounds in this mass window on the mixing with active neutrinos, see [1,2] for reviews. In particular, intensity frontier experiments can put relevant bounds in the mass-mixing plane, see e.g. [11–15]. On the other hand, the case of a sterile neutrino dipole has been considered, in the limit of negligible active-sterile mixing, in a few papers [16–22].

Here we will analyse extensively the interplay between the dipole and the mixing, and determine the sensitivity of upcoming experiments when both are taken into account. As we will see, the dipole operator between two sterile neutrinos also generates a dipole between active and sterile neutrinos, as a consequence of the mixing. Such *dipole portal* between SM neutrinos and sterile neutrinos has been extensively studied in the literature [23–42].

The dipole portal is usually postulated, without connection to active-sterile mixing. In this article, instead, we will carefully take into account the correlations between the two, which will play an important role in the phenomenology. In addition, we point out that, as long as sterile neutrinos have a non-vanishing mixing with active neutrinos, an irreducible active-sterile dipole is induced by one-loop electroweak corrections [43,44]. We will include this contribution in our analysis, and show that it is important in some regions of parameter space, where the photon signal can be observed even in the absence of higher-dimensional dipole operators.

We start in section 2 with a careful analysis of the minimal model, involving two sterile neutrinos $N_{1,2}$. We study the symmetries controlling the relative size of the sterile mass, dipole, and mass splitting. We then perform an explicit, analytic computation of the couplings of the neutrino mass eigenstates. Finally, we generalise to the case of three lepton flavours and review the flavour-dependent bounds on the active-sterile mixing.

We then move to the analysis of sterile neutrino production and detection in section 3. We compute the number of steriles produced from meson decays in upcoming beam-dump experiments (NA62, SHiP) and proton-proton collisions (FASER2), as well as the sterile decay widths in all possible channels. This allows us to identify the regions where the dipole dominates over the mixing, or vice versa. We then simulate events according to the various detector geometries, to compute how many sterile neutrinos decay into a photon above threshold in the detector. This leads us to determine the sensitivity of each experiment as a function of the sterile neutrino masses and the size of the dipole coupling, taking into account the upper bounds on the active-sterile mixing. Constraints from a variety of past experiments are also presented.

We summarise our main results in section 4. Appendix A details the computation of the dipole-induced $N$ decay widths, and clarifies the limit in which $N_1$ and $N_2$ combine into one Dirac neutrino. Appendix B provides the analytic expressions for the neutrino dipole induced by electroweak loops. Appendix C collects the relevant formulas for meson and sterile-neutrino decay widths in the various channels, and compares them quantitatively.

## 2 Model: Two sterile neutrinos with one dipole

Let us add to the Standard Model (SM) two Weyl fermions $N'_{1,2}$ neutral with respect to SM gauge symmetries, i.e. two sterile neutrinos. The relevant Lagrangian is

$$
\begin{aligned}
\mathcal{L}_N = {}& iN_1'^{\dagger}\bar{\sigma}^{\mu}\partial_{\mu}N_1' + iN_2'^{\dagger}\bar{\sigma}^{\mu}\partial_{\mu}N_2' - \left(MN_1'N_2' + \frac{\mu_1}{2}N_1'N_1' + \frac{\mu_2}{2}N_2'N_2' + \text{h.c.}\right) \\
& - \left[\tilde{H}^{\dagger}L'\left(Y_N^1 N_1' + Y_N^2 N_2'\right) + \text{h.c.}\right] + \left(d\, N_1'\sigma^{\mu\nu}N_2'B_{\mu\nu} + \text{h.c.}\right),
\end{aligned}
\tag{1}
$$

where we use the two-component spinor notation, with the conventions of Ref. [45], and we adopt a primed notation for the fermion fields, as we will reserve an unprimed notation

for the mass eigenstates. Beside the $N'_{1,2}$ kinetic and mass terms, the Lagrangian contains their Yukawa couplings to the SM Higgs doublet $H$ and lepton doublet $L'$ (for the moment we neglect different lepton flavours), as well as a dipole coupling to the hypercharge field strength $B_{\mu\nu}$. The Wilson coefficient of the dimension-five dipole operator has dimension -1 and is complex in general: we parametrise it as $d = |d|e^{i\xi}$. Note that two is the minimum number of sterile states to introduce a dipole interaction, since the latter is antisymmetric under fermion exchange.

The Lagrangian in Eq. (1) contains all sterile-neutrino operators allowed by symmetries up to dimension five, except for the operators $N'_i N'_j H^\dagger H = N'_i N'_j (v + h)^2/2$, where $v \simeq 246$ GeV is the Higgs vacuum expectation value and $h$ the physical Higgs boson. These operators shift the sterile neutrino Majorana masses and may affect Higgs physics, but they are irrelevant for our purposes and will be neglected in the following. Recall that the SM effective field theory contains a single dimension-five, Weinberg operator, $(HL')(HL')$: in general, SM neutrinos may receive a mass both from the latter and from the mixing with $N'_{1,2}$, as discussed later. We will neglect operators with dimension larger than five.

Since dipole operators can only be generated at loop level, we expect the Wilson coefficient $d$ to satisfy the following power counting:

$$d \simeq \frac{g'}{16\pi^2} \frac{g_\star^2}{m_\star}, \tag{2}$$

where $g'$ is the gauge coupling associated with hypercharge, $g_\star$ is a typical coupling between $N'_{1,2}$ and the ultraviolet (UV) physics that generates the effective operator, while $m_\star$ represents a typical mass in the UV theory. In weakly coupled theories we expect $g_\star \lesssim 1$, while in strongly coupled theories the coupling can be as large as $g_\star \sim 4\pi$, in such a way to (partially) compensate the loop suppression.

## 2.1 Symmetries

It is interesting to analyse the symmetries of the Lagrangian. For two sterile neutrino species $N'_{1,2}$, the kinetic term has a global symmetry $SU(2)_N \times U(1)_N$, acting on the doublet $\boldsymbol{N}' \equiv (N'_1 N'_2)^T$ as

$$SU(2)_N: \quad \boldsymbol{N}' \to V_N \boldsymbol{N}', \qquad U(1)_N: \quad \boldsymbol{N}' \to e^{i\alpha_N} \boldsymbol{N}'. \tag{3}$$

Let us observe that, because of the antisymmetry of the $\sigma^{\mu\nu}$ spinor structure

$$N'_1 \sigma^{\mu\nu} N'_2 = \frac{1}{2} \left( N'_1 \sigma^{\mu\nu} N'_2 - N'_2 \sigma^{\mu\nu} N'_1 \right) = \frac{1}{2} \boldsymbol{N}'_a \sigma^{\mu\nu} \epsilon^{ab} \boldsymbol{N}'_b, \tag{4}$$

where the antisymmetric Levi-Civita tensor $\epsilon$ acts on the family index of the sterile neutrinos. Therefore, the dipole operator breaks the $U(1)_N$ symmetry, but it preserves $SU(2)_N$ with $\boldsymbol{N}'$ transforming as a doublet.

In contrast, the sterile neutrino mass matrix

$$\mathcal{M} = \begin{pmatrix} \mu_1 & M \\ M & \mu_2 \end{pmatrix} = \begin{pmatrix} -m_1 + im_2 & m_3 \\ m_3 & m_1 + im_2 \end{pmatrix}, \tag{5}$$

can break both $U(1)_N$ and $SU(2)_N$. More precisely, the spurion $\mathcal{M}^{ab} \equiv \sum_A m_A (\sigma_A \epsilon)^{ab}$ transforms in the triplet representation of $SU(2)_N$. In the case of real $m_A$, $SU(2)_N$ is broken to a $U(1)'_N$ subgroup, and one can check that the two sterile mass eigenstates are degenerate, with $M_1 = M_2 = (\sum_A m_A m_A)^{1/2}$. This corresponds to a Dirac neutrino with a conserved $U(1)'_N$ charge. Without loss of generality, one can take the real-triplet spurion in the third-component

direction, $m_1 = m_2 = 0$ and $m_3 = M$, which preserves the $U(1)'_N$ subgroup generated by $\sigma_3$, with charges $q(N'_1) = +1$ and $q(N'_2) = -1$.

In general, $m_A$ are complex and can break $SU(2)_N$ fully: in this case there are two non-degenerate Majorana fermions, with masses

$$M^2_{1,2} = \sum_A m_A m_A^* \pm 2 \left( [\text{Im}(m_1 m_2^*)]^2 + [\text{Im}(m_2 m_3^*)]^2 + [\text{Im}(m_3 m_1^*)]^2 \right)^{1/2} . \tag{6}$$

It is technically natural to have a small mass difference $\Delta M \equiv M_2 - M_1$, as a $U(1)'_N$ symmetry is restored in the limit $\Delta M \to 0$.

Let us now introduce the SM lepton doublet $L'$. As long as the Yukawa couplings $Y^{1,2}_N$ between $L'$ and the sterile neutrinos $N'_{1,2}$ are set to zero, there is no active-sterile mixing, and the SM neutrino $\nu'$ remains massless. However, there is an alternative, less trivial way to keep a neutrino light while allowing for active-sterile mixing: it is sufficient to generalise the $U(1)'_N$ symmetry introduced above, by assigning a charge $q(L') = +1$. This symmetry can be considered as a lepton number symmetry, $U(1)_L$, and it is preserved for $Y^1_N \to 0$ together with $\mu_{1,2} \to 0$. In this limit the spectrum is formed by a massless, mostly active neutrino $\nu$, and a massive, mostly sterile Dirac fermion $N$. When $U(1)_L$ is explicitly broken by $Y^1_N$ and/or $\mu_{1,2}$, the spectrum is formed by three massive Majorana fermions: a light mass eigenstate, $\nu$, and two heavy ones, $N_{1,2}$.

Given this pattern of symmetry breaking, we conclude that it is technically natural to take the dipole coefficient $d$ as large as allowed by perturbativity, while the sterile neutrino mass scale $M$ is kept small as desired, since a symmetry $SU(2)_N$ is recovered in the limit in which it vanishes. In addition, for a given $M$, it is technically natural to have an even smaller sterile mass difference $\Delta M$ as well as a tiny active neutrino mass $m_\nu$, since $U(1)_L$ is recovered in the limit in which they vanish. We note in passing that this mass splitting is a crucial parameter to realise leptogenesis close to the GeV scale, see e.g. [46] for a recent analysis. Also, a small, non-zero $\Delta M$ can be exploited to obtain evidence for lepton-number violation, see e.g. [47] for a study with the SHiP experiment.

## 2.2 Mass diagonalisation

The primed neutrino fields can be expressed as a function of the mass eigenstate neutrinos via

$$\begin{pmatrix} \nu' \\ N'_1 \\ N'_2 \end{pmatrix} = U \begin{pmatrix} \nu \\ N_1 \\ N_2 \end{pmatrix} , \tag{7}$$

where the unitary matrix $U$ satisfies the condition $U^T \mathcal{M} U = \mathcal{M}_{diag}$, with $\mathcal{M}_{diag}$ the diagonal matrix of mass eigenvalues and

$$\mathcal{M} = \begin{pmatrix} 0 & Y^1_N v & Y^2_N v \\ Y^1_N v & \mu_1 & M \\ Y^2_N v & M & \mu_2 \end{pmatrix} = \left( \begin{array}{c|c} 0 & m_D^T \\ \hline m_D & \mathcal{M} \end{array} \right) , \tag{8}$$

where $m_D^T \equiv Y_N v$ is a 2-component row vector, the $2 \times 2$ matrix $\mathcal{M}$ has been already defined in Eq. (5), and we used $\langle H \rangle = v \simeq 174$ GeV for the Higgs vev. Using the unitarity of $U$, we can write

$$U = \begin{pmatrix} e^{i\alpha} \sqrt{1 - \boldsymbol{\theta}^\dagger \boldsymbol{\theta}} & \boldsymbol{\theta}^\dagger \\ -\dfrac{e^{i\alpha} \mathcal{U} \boldsymbol{\theta}}{\sqrt{1 - \boldsymbol{\theta}^\dagger \boldsymbol{\theta}}} & \mathcal{U} \end{pmatrix} , \tag{9}$$

where $\alpha$ is a real number, $\boldsymbol{\theta} = (\theta_1\,\theta_2)^T$ is a 2-component complex vector, representing the mixing between the active and sterile neutrinos, and $\mathcal{U}$ is a $2 \times 2$ matrix that satisfies

$$\mathcal{U}^\dagger\mathcal{U} = \mathbb{1} - \boldsymbol{\theta}\boldsymbol{\theta}^\dagger = V_\theta \begin{pmatrix} 1 - \boldsymbol{\theta}^\dagger\boldsymbol{\theta} & 0 \\ 0 & 1 \end{pmatrix} V_\theta^\dagger \,, \tag{10}$$

with $V_\theta$ the unitary matrix that diagonalises $\mathcal{U}^\dagger\mathcal{U}$. Eq. (10) fixes $\mathcal{U}$ as a function of $\boldsymbol{\theta}$, up to an arbitrary unitary rotation $V$ on the left,

$$\mathcal{U} = V \begin{pmatrix} \sqrt{1 - \boldsymbol{\theta}^\dagger\boldsymbol{\theta}} & 0 \\ 0 & 1 \end{pmatrix} V_\theta^\dagger \,, \qquad V_\theta = \frac{1}{\sqrt{\boldsymbol{\theta}^\dagger\boldsymbol{\theta}}} \begin{pmatrix} \theta_1 & -i\theta_2^* \\ \theta_2 & i\theta_1^* \end{pmatrix} \,. \tag{11}$$

The matrix $V$ can be parameterised e.g. as $V = diag(e^{i\beta}, e^{i\gamma}) R_3 \, diag(e^{i\delta}, e^{-i\delta})$, with $R_3$ a real orthogonal rotation by an angle $\theta_3$. After a phase redefinition $e^{i\alpha}\boldsymbol{\theta} \to \boldsymbol{\theta}$, the phases $\alpha, \beta, \gamma$ can be removed by a phase redefinition of the fields $\nu', N_1', N_2'$, therefore we will drop them in the following. In summary, the six physical parameters in the diagonalisation matrix $U$ are $|\theta_{1,2}|$, $\arg(\theta_{1,2})$, $\theta_3$ and $\delta$.

In terms of mass eigenstates, the dipole interaction term can be written as

$$\mathcal{L}_{\text{dipole}} = \frac{d}{2} \left( \boldsymbol{N}^T - \frac{\nu\boldsymbol{\theta}^T}{\sqrt{1 - \boldsymbol{\theta}^\dagger\boldsymbol{\theta}}} \right) \mathcal{U}^T \sigma^{\mu\nu} \epsilon \, \mathcal{U} \left( \boldsymbol{N} - \frac{\boldsymbol{\theta}\nu}{\sqrt{1 - \boldsymbol{\theta}^\dagger\boldsymbol{\theta}}} \right) B_{\mu\nu} + \text{h.c.} \,, \tag{12}$$

where $\boldsymbol{N} \equiv (N_1\,N_2)^T$. Since $\mathcal{U}^T\epsilon\mathcal{U} = \det\mathcal{U} \cdot \epsilon = e^{i\phi}\sqrt{1 - \boldsymbol{\theta}^\dagger\boldsymbol{\theta}} \cdot \epsilon$, the dipole interaction becomes

$$\mathcal{L}_{\text{dipole}} = d\, e^{i\phi} \left( \sqrt{1 - \boldsymbol{\theta}^\dagger\boldsymbol{\theta}}\, N_1\sigma^{\mu\nu}N_2 B_{\mu\nu} - \theta_2 N_1\sigma^{\mu\nu}\nu B_{\mu\nu} + \theta_1 N_2\sigma^{\mu\nu}\nu B_{\mu\nu} \right) + \text{h.c.} \tag{13}$$

Note that the active-active dipole remains exactly zero by antisymmetry. Since the active-sterile mixing angles $\theta_i$ are small, we will sometime neglect terms of order $\theta_i\theta_j$, i.e. drop the square root in the coefficient of the $N_1 - N_2$ dipole.

The seesaw mass matrix of Eq. (8) can be block-diagonalised perturbatively in the matrix $\epsilon \equiv \mathcal{M}^{-1}m_D$ [48]. To leading order, we have

$$\mathcal{U}\boldsymbol{\theta} \simeq \mathcal{M}^{-1}m_D \,. \tag{14}$$

As usual, the light (mostly active) neutrino mass is given by the seesaw formula,

$$m_\nu \simeq -m_D^T\mathcal{M}^{-1}m_D \simeq -\boldsymbol{\theta}^T\mathcal{U}^T\mathcal{M}\,\mathcal{U}\boldsymbol{\theta} \,, \tag{15}$$

which shows that $m_\nu$ is suppressed by terms of order $\theta_i\theta_j$ with respect to the sterile neutrino mass scale. The masses $M_{1,2}$ of the heavy (mostly sterile) neutrinos are given by Eq. (6), up to corrections of order $\theta_i\theta_j$.

It is interesting to spell out the $U(1)_L$ conserving limit, where $m_{D1} = \mu_1 = \mu_2 = 0$. In this case the diagonal mass matrix is $\mathcal{M}_{diag} = diag(0, M_D, M_D)$, with a Dirac mass given by $M_D = \sqrt{m_{D2}^2 + M^2}$, and the active-sterile mixing parameters are equal to

$$\boldsymbol{\theta} = \frac{1}{\sqrt{2}} \begin{pmatrix} is \\ s \end{pmatrix} \,, \qquad \mathcal{U} = \frac{1}{\sqrt{2}} \begin{pmatrix} -ic & c \\ i & 1 \end{pmatrix} \,, \tag{16}$$

with $s \equiv m_{D2}/\sqrt{m_{D2}^2 + M^2}$ and $c \equiv M/\sqrt{m_{D2}^2 + M^2}$. Even though this mixing does not contribute to $m_\nu$, it does modify the $\nu$ couplings to the $W$ and $Z$ bosons: experimental bounds

require roughly $s^2 \lesssim 10^{-3}$ [49]. Introducing four-component spinors for the Dirac mass eigenstate and the massless neutrino,

$$N_D = \begin{pmatrix} cN_1' + s\nu' \\ N_2'^\dagger \end{pmatrix}, \qquad \nu_L = \begin{pmatrix} -sN_1' + c\nu' \\ 0 \end{pmatrix}, \tag{17}$$

the dipole interaction reads

$$\mathcal{L}_{\text{dipole}} = -\frac{1}{2} d\, c\, \overline{N_D} \Sigma^{\mu\nu} P_L N_D B_{\mu\nu} + \frac{1}{2} d\, s\, \overline{N_D} \Sigma^{\mu\nu} \nu_L B_{\mu\nu} + \text{h.c.}, \tag{18}$$

where $\Sigma^{\mu\nu} = i[\gamma^\mu, \gamma^\nu]/2$ [45].

Finally, let us consider one particular departure from the $U(1)_L$ limit, which amounts to split $N_D$ into a pair of non-degenerate Majorana fermions, while keeping $m_\nu$ vanishing. To this purpose, let us choose $\mathcal{M}_{diag} \equiv diag(m_\nu, M_1, M_2) = diag[0, M_s(1-\delta), M_s(1+\delta)]$. In this case $\mathcal{M}_{11} = 0$ implies $\theta_1^2(1-\delta) + \theta_2^2(1+\delta) = 0$, so for the mixing one can take

$$\boldsymbol{\theta} = \frac{1}{\sqrt{2}} \begin{pmatrix} is\sqrt{1+\delta} \\ s\sqrt{1-\delta} \end{pmatrix}, \qquad \mathcal{U} = \frac{1}{\sqrt{2}} \begin{pmatrix} -ic\sqrt{1+\delta} & c\sqrt{1-\delta} \\ i\sqrt{1-\delta} & \sqrt{1+\delta} \end{pmatrix}, \tag{19}$$

which generalises Eq. (16) with the same definition for $s$ and $c$, and $\mathcal{U}$ was determined from $\boldsymbol{\theta}$ by using Eq. (11) and setting $V$ to the identity. The sterile masses are related to the Dirac mass by $M_s = M_D/\sqrt{1-\delta^2}$. This particular scenario corresponds to the $U(1)_L$-breaking parameter $\mu_2 = 2\delta M_s$, while keeping $m_{D1} = \mu_1 = 0$. This choice is neither generic nor justified by a symmetry, rather it corresponds to a convenient slice of the allowed parameter space, which is sufficient to study the phenomenology as a function of the sterile neutrino mass splitting $\delta$, defined by

$$\delta \equiv \frac{M_2 - M_1}{M_2 + M_1}. \tag{20}$$

## 2.3 Other sources of the active-sterile dipole

Eq. (13) shows that an active-sterile dipole emerges from the dim-five dipole operator of the sterile neutrinos, via the active-sterile mixing.

We observe that an active-sterile dipole may be generated, alternatively, by introducing a dim-six operator, $d_6 N\sigma^{\mu\nu}(LH)B_{\mu\nu}$, with a Wilson coefficient $d_6$ which, in general, is unrelated to the active-sterile mixing. We assume that this contribution is negligible with respect to the one that appears Eq. (13). The $d_6$ contribution is actually subdominant in a large class of UV theories of flavour (e.g. partial compositeness), where the size of $d_6$ is related to the size of $d$ and $\theta \simeq Y_N v/M$, as the associated operators involve the same fields. In these theories, combining the estimate for $d$ in Eq. (2) with analogous power-counting estimates for $d_6$ and $Y_N$, we find a ratio $(d_6 \cdot v)/(d \cdot \theta) \simeq M/m_*$. Such ratio is smaller than one as long as sterile neutrinos are lighter than the UV cutoff. We thus neglect $d_6$ in the following.

Active-sterile dipoles are also generated by electroweak (EW) interactions at one loop. In particular, a dipole coupling to the photon emerges from loops involving a charged lepton and a $W$ boson, as detailed in Appendix B. The result is

$$d_{N_k\nu}^{EW} \simeq -\frac{3eG_F}{8\sqrt{2}\pi^2} M_k \theta_k^*, \quad k = 1, 2. \tag{21}$$

In general, this coefficient should be added to the new physics (NP) contribution from Eq. (13), $d_{N_1\nu}^{NP} = -d\, e^{i\phi}\, \theta_2 \cos\theta_w$ and $d_{N_2\nu}^{NP} = d\, e^{i\phi}\, \theta_1 \cos\theta_w$, where $\theta_w$ is the weak mixing angle connecting hypercharge to electric charge. Inserting the NDA estimate for $d$ given in Eq. (2), and

assuming $|\theta_1| \simeq |\theta_2|$, the relative size of the two contributions can be written as

$$\frac{|d_{N_k \nu}^{EW}|}{|d_{N_k \nu}^{NP}|} \simeq 0.05 \left(\frac{1}{g_*}\right)^2 \left(\frac{m_*}{\text{TeV}}\right) \left(\frac{M_k}{\text{GeV}}\right). \tag{22}$$

Therefore, the EW contribution becomes relatively more important as $N_k$ become heavier, and as the NP states decouple from $N_k$. Whenever relevant for our analysis, we will take into account the total electromagnetic dipole coefficient, $d^{em} \equiv d^{NP} + d^{EW}$.

## 2.4 Some considerations on lepton flavour

Up to this point we neglected lepton flavour, assuming a single SM lepton doublet $L = (\nu \ e)^T$. However, the experimental bounds on charged-lepton transitions, as well as the measured neutrino-oscillation parameters, have a strong flavour dependence. Let us therefore introduce the three different flavours of lepton doublets, $L_\alpha$ for $\alpha = e, \mu, \tau$. Then, the active-sterile mixing is promoted to a $2 \times 3$ matrix with entries $\theta_{i\alpha}$.

Let us stick to the $U(1)_L$ conserving limit, which guarantees vanishingly small neutrino masses. In this limit $\mu_{1,2} = 0$ and $Y_N^{1\alpha} = 0$, therefore the neutrino Dirac mass matrix $m_D$ and the sterile mass matrix $\mathcal{M}$, introduced in Eq. (8), read

$$m_D = \begin{pmatrix} 0 & 0 & 0 \\ m_{2e} & m_{2\mu} & m_{2\tau} \end{pmatrix}, \qquad \mathcal{M} = \begin{pmatrix} 0 & M \\ M & 0 \end{pmatrix}, \tag{23}$$

for $m_{2\alpha} \equiv Y_N^{2\alpha} v$. The diagonalised mass matrix is $\mathcal{M}_{diag} = \text{diag}(0, 0, 0, M_D, M_D)$, with $M_D = \sqrt{m_{2e}^2 + m_{2\mu}^2 + m_{2\tau}^2 + M^2}$, and the diagonalisation matrix $U$, defined by Eq. (7), takes the form

$$U = \begin{pmatrix} c_e & 0 & 0 & s_e & 0 \\ -s_e s_\mu & c_\mu & 0 & c_e s_\mu & 0 \\ -s_e c_\mu s_\tau & -s_\mu s_\tau & c_\tau & c_e c_\mu s_\tau & 0 \\ -s_e c_\mu c_\tau & -s_\mu c_\tau & -s_\tau & c_e c_\mu c_\tau & 0 \\ 0 & 0 & 0 & 0 & 1 \end{pmatrix} \begin{pmatrix} 1 & 0 & 0 & 0 & 0 \\ 0 & 1 & 0 & 0 & 0 \\ 0 & 0 & 1 & 0 & 0 \\ 0 & 0 & 0 & -i/\sqrt{2} & 1/\sqrt{2} \\ 0 & 0 & 0 & i/\sqrt{2} & 1/\sqrt{2} \end{pmatrix}, \tag{24}$$

where $s_\alpha \equiv \sin\theta_\alpha$, $c_\alpha \equiv \cos\theta_\alpha$, and

$$s_e \equiv \frac{m_{2e}}{\sqrt{m_{2e}^2 + m_{2\mu}^2 + m_{2\tau}^2 + M^2}}, \qquad s_\mu \equiv \frac{m_{2\mu}}{\sqrt{m_{2\mu}^2 + m_{2\tau}^2 + M^2}}, \qquad s_\tau \equiv \frac{m_{2\tau}}{\sqrt{m_{2\tau}^2 + M^2}}. \tag{25}$$

This corresponds to

$$\boldsymbol{\theta} = \frac{1}{\sqrt{2}} \begin{pmatrix} is_e & ic_e s_\mu & ic_e c_\mu s_\tau \\ s_e & c_e s_\mu & c_e c_\mu s_\tau \end{pmatrix}, \qquad \mathcal{U} = \frac{1}{\sqrt{2}} \begin{pmatrix} -ic_e c_\mu c_\tau & c_e c_\mu c_\tau \\ i & 1 \end{pmatrix}. \tag{26}$$

It is easy to check that, in the limit where only one $m_{2\alpha}$ is non-zero, one reduces to the single flavour case of Eq. (16). Note that, since the three light neutrinos are massless in this $U(1)_L$-conserving limit, the matrix $U$ is determined only up to a block-diagonal transformation from the right-hand side, of the form $\tilde{V} = diag(V', \mathbb{1})$, with $V'$ an arbitrary $3 \times 3$ unitary matrix and $\mathbb{1}$ the $2 \times 2$ identity matrix.

Let us discuss the constraints on the various flavour mixing parameters. To begin with, consider sterile neutrinos with mass above the electroweak scale, which can be probed only

indirectly via their contribution to SM higher-dimensional operators. In this case, lepton-flavour conserving observables (mainly, corrections to $Z$ and $W$ couplings, and to the muon decay) imply [49]

$$s_e^2 \lesssim 10^{-3}, \qquad s_\mu^2 \lesssim 10^{-3}, \qquad s_\tau^2 \lesssim 3 \times 10^{-3}. \tag{27}$$

In addition, lepton-flavour violating $\mu$-to-$e$ transitions imply more severe bounds, when both mixing angles are non-vanishing [49],

$$\text{if } s_e \simeq s_\mu, \quad \text{then } s_{e,\mu}^2 \lesssim 10^{-5} \, [10^{-7}], \tag{28}$$

where the number in bracket indicates the sensitivity of the next generation experiments. Current bounds on $\tau$ flavour-violating observables are not more constraining then Eq. (27), but they might be in the near future. Note that most of these indirect constraints come from low-energy observables, at the $\mu$ ($\tau$) mass scale, therefore they apply also to sterile neutrinos as light as a GeV (a few GeVs).

However, if sterile neutrinos are at the electroweak scale or below, direct searches may imply additional constraints on the active-sterile mixing angles. A compilation of such constraints can be found e.g. in [1,2]. Assuming mixing with a single lepton family at a time, the current bounds can be summarised, very roughly, as follows:

$$\begin{aligned}
&\text{for } 2 \text{ GeV} \lesssim M_D \lesssim 80 \text{ GeV}, && s_{e,\mu}^2 \lesssim 10^{-5}, && s_\tau^2 \lesssim 10^{-5}, \\
&\text{for } 0.5 \text{ GeV} \lesssim M_D \lesssim 2 \text{ GeV}, && s_{e,\mu}^2 \lesssim 10^{-7}, && s_\tau^2 \lesssim 10^{-6}, \\
&\text{for } 0.2 \text{ GeV} \lesssim M_D \lesssim 0.5 \text{ GeV}, && s_{e,\mu}^2 \lesssim 10^{-9}, && s_\tau^2 \lesssim 10^{-5}.
\end{aligned} \tag{29}$$

We refer to [1,2] for more precise numbers and a detailed discussion of the various searches. Note that these bounds are derived assuming sterile neutrinos with no dipole interactions: as we will see, the dipole modifies the sterile neutrino production and decay channels, potentially shifting these constraints on the mixing angles. Another potentially strong bound comes from Big Bang Nucleosynthesis [1,2]: the lifetime of sterile neutrinos should be smaller than about one second. This requirement, combined with the limits of Eq. (29), excludes all masses $M_D \lesssim 0.5$ GeV (when $s_{e,\mu}^2 \neq 0$) or $M_D \lesssim 0.1$ GeV ($s_\tau^2 \neq 0$). Also this bound can be modified by the presence of the dipole operator, as we will see in Sec. 3.2.

The sterile-neutrino dipole operator reads, in the mass basis,

$$\begin{aligned}
\mathcal{L}_{\text{dipole}} = &-\frac{1}{2} \, d \, c_e c_\mu c_\tau \, \overline{N_D} \Sigma^{\mu\nu} P_L N_D B_{\mu\nu} \\
&+\frac{1}{2} \, d \, \overline{N_D} \Sigma^{\mu\nu} \big( s_\tau \, \nu_{L\tau} + c_\tau s_\mu \, \nu_{L\mu} + c_\tau c_\mu s_e \, \nu_{Le} \big) B_{\mu\nu} + \text{h.c.},
\end{aligned} \tag{30}$$

where $\nu_{Le,\mu,\tau}$ are the three massless eigenstates in the flavour basis: their relation to neutrino mass eigenstates has yet to be determined, by the lepton-number-violating parameters which generate their masses. Note that $\nu_{Le,\mu,\tau}$ have no dipole interaction among each other, since those would violate lepton number by two units. Indeed, a departure from the $U(1)_L$ limit could also induce transitional dipole moments among the three light Majorana neutrinos, which are subject to strong experimental constraints [50].

Departures from the $U(1)_L$-conserving limit are also constrained by the smallness of the light neutrino Majorana mass matrix $m_\nu$, whose entries should not exceed $\sim 0.1$ eV. By generalising what we showed for the one-flavour case at the end of section 2.2, one possible breaking of lepton number amounts to take $\mathcal{M}_{diag} = diag[0,0,0,(1-\delta)M_s,(1+\delta)M_s]$, thus splitting the two sterile masses while keeping $m_\nu = 0$. The mixing matrix $U$ is minimally modified with respect to Eq. (24), by the replacement of the $2 \times 2$ block in the second factor,

$$\frac{1}{\sqrt{2}} \begin{pmatrix} -i & 1 \\ i & 1 \end{pmatrix} \quad \Rightarrow \quad \frac{1}{\sqrt{2}} \begin{pmatrix} -i\sqrt{1+\delta} & \sqrt{1-\delta} \\ i\sqrt{1-\delta} & \sqrt{1+\delta} \end{pmatrix}. \tag{31}$$

The corresponding modification to Eq. (23) amounts to a non-zero 22-entry in $\mathcal{M}$, given by $\mu_2 = 2\delta M_s$, with $M_s = M_D/\sqrt{1-\delta^2}$. Note that in this scenario one has

$$|\theta_{2\alpha}| = \sqrt{\frac{1-\delta}{1+\delta}}\,|\theta_{1\alpha}|, \quad \alpha = e, \mu, \tau. \tag{32}$$

This particular $U(1)_L$ breaking has the additional advantage of not inducing transitional dipole moments among the light neutrinos. Indeed, we started from $dN_1'\sigma^{\mu\nu}N_2'B_{\mu\nu}$, but $N_2'$ does not contain any light neutrino component, because $U_{5i} = 0$ for $i = 1, 2, 3$. Explicitly,

$$\begin{aligned}
\mathcal{L}_{\text{dipole}} = &-i\,d\,c_e c_\mu c_\tau\, N_1\sigma^{\mu\nu}N_2 B_{\mu\nu} \\
&+ \frac{i}{\sqrt{2}}\,d\,\sqrt{1-\delta}\,N_1\sigma^{\mu\nu}\big(s_\tau\nu_\tau + c_\tau s_\mu\nu_\mu + c_\tau c_\mu s_e\nu_e\big)B_{\mu\nu} \\
&+ \frac{1}{\sqrt{2}}\,d\,\sqrt{1+\delta}\,N_2\sigma^{\mu\nu}\big(s_\tau\nu_\tau + c_\tau s_\mu\nu_\mu + c_\tau c_\mu s_e\nu_e\big)B_{\mu\nu} + \text{h.c.}
\end{aligned} \tag{33}$$

A more general $U(1)_L$ breaking can contribute to the $3\times 3$ light neutrino mass matrix $m_\nu$. Indeed, the mixing with the two sterile neutrinos $N_{1,2}'$ could be sufficient to fit all neutrino oscillation data, which determine accurately the mass differences and mixing angles among the three flavours of light neutrinos. In general, $m_\nu$ might receive additional contributions from other UV sources, such as a Weinberg operator or additional, heavier sterile neutrinos. However, if one insists that the $N_{1,2}'$ contribution dominates, then the active-sterile mixing can be further constrained. Introducing the 'relative' mixing angles $\hat{s}_\alpha^2 \equiv s_\alpha^2/(\sum_\beta s_\beta^2)$, and requiring to reproduce oscillation data with only two sterile neutrinos, one roughly finds [51],

$$\begin{array}{llll}
\text{normal ordering } (m_1 < m_2 < m_3): & \hat{s}_e^2 \lesssim 0.1, & 0.25 \lesssim \hat{s}_\mu^2 \lesssim 0.85, & \hat{s}_\tau^2 \simeq 1-\hat{s}_\mu^2, \\
\text{inverted ordering } (m_3 < m_1 < m_2): & 0.05 \lesssim \hat{s}_e^2 \lesssim 0.95, & \hat{s}_\mu^2 \simeq \hat{s}_\tau^2 \simeq 0.5(1-\hat{s}_e^2),
\end{array} \tag{34}$$

where $m_{1,2,3}$ are the light neutrino mass eigenvalues. In other words, the relative sizes of $s_{e,\mu,\tau}$ are no longer independent, therefore the upper bounds on each $s_\alpha^2$, listed earlier in this section, now may translate into bounds on the other flavours, according to Eq. (34).

## 3 Phenomenology at the intensity frontier

### 3.1 Sterile neutrino production and decays

The decays $N_2 \to N_1\gamma$ and $N_k \to \nu_\alpha\gamma$ are central in our phenomenological analysis. The $N_2$-to-$N_1$ radiative decay is driven by the dimension-five dipole coefficient, from the first line of Eq. (33), $d_{N_2 N_1}^{em} \simeq i\,d\cos\theta_w$ (the EW correction is negligible, being suppressed by two powers of the active-sterile mixing). On the other hand, the $N_k$-to-$\nu_\alpha$ radiative decay is controlled by $d_{N_k\nu_\alpha}^{em} = d_{N_k\nu_\alpha}^{NP} + d_{N_k\nu_\alpha}^{EW}$, where the NP contribution can be read off the second and third lines of Eq. (33), while the EW contribution is given by Eq. (21) with the obvious replacement $\theta_k \to \theta_{k\alpha}$.

At leading order in the mixing angles $\theta_{k\alpha}$, the decay widths are given by

$$\Gamma(N_2 \to N_1\gamma) = \frac{|d|^2\cos^2\theta_w}{8\pi}M_2^3\left(1 - \frac{M_1^2}{M_2^2}\right)^3,$$

$$\Gamma(N_2 \to \nu_\alpha\gamma) = \frac{1}{8\pi}\left|d\cos\theta_w\sqrt{\frac{1+\delta}{1-\delta}} - \frac{3eG_F}{8\sqrt{2}\pi^2}M_2\right|^2|\theta_{2\alpha}|^2 M_2^3, \tag{35}$$

$$\Gamma(N_1 \to \nu_\alpha\gamma) = \frac{1}{8\pi}\left|d\cos\theta_w\sqrt{\frac{1-\delta}{1+\delta}} + \frac{3eG_F}{8\sqrt{2}\pi^2}M_1\right|^2|\theta_{1\alpha}|^2 M_1^3.$$

In numerical applications, for definiteness we will always take $d$ purely imaginary. This choice makes the interference between the NP and EW dipole vanish. Note that, no matter what the phase of $d$ is, the interference cannot be destructive at the same time for $N_1$ and $N_2$. The limit in which $\delta \to 0$ and the two Majorana fermions $N_{1,2}$ combine to form a unique Dirac particle is clarified in App. A.

In general, all these decay widths are expected to be small, i.e. both $N_{1,2}$ are expected to be long-lived. Various factors point to this conclusion. First, we expect $|d|M_k \ll 1$ in order to be inside the range of validity of the effective theory we are considering. Second, for GeV scale sterile neutrinos, the EW contribution is suppressed by $G_F M_k^2 \ll 1$, and in addition it is loop suppressed. Third, following our discussion of symmetries in Sec. 2.1, we expect $M_1$ and $M_2$ to be relatively close in mass, since their mass splitting is due to explicit $U(1)_L$ breaking. Finally, the mixing angles $|\theta_{k\alpha}|$ are also small because of experimental limits, as discussed in Sec. 2.4. Inspecting Eq. (35), we see that $\Gamma(N_2 \to N_1 \gamma)$ is suppressed by the smallness of the dipole coefficient and by the small mass splitting; the widths $\Gamma(N_{1,2} \to \nu_\alpha \gamma)$, on the other hand, are suppressed by the smallness of the dipole coefficient, of the EW loops, and of the mixing angles. This leads to the conclusion that, if produced in sufficient numbers, $N_{1,2}$ decays can give interesting signals at experiments that may exploit their long-lived nature, i.e. experiments at the intensity frontier.[2]

Once the kinematics (i.e. $M_{1,2}$) is fixed, the phenomenology is completely determined by the dipole coefficient $d$ and by the mixing angles $\theta_{k\alpha}$. Different regions in parameter space will have different phenomenology, depending on the relative size of dipole and mixing. For instance, we can have the dipole (or the mixing) dominating both $N_{1,2}$ production and decays, while in other regions the dipole may dominate production and the mixing may dominate decays, or viceversa. The shape and extension of these regions will depend not only on the parameters $d$ and $\theta_{k\alpha}$, but also on the experiment considered, since the number of sterile states produced is experiment-dependent. To estimate the region in which the dipole dominates, we will consider, as an example, three experiments at the intensity frontier, the same for which we will later compute the sensitivities: SHiP [8], NA62 [9] and FASER2 [10]. All these experiments consist in a proton beam colliding with either a target (NA62 and SHiP, that use or will use the 400 GeV SPS proton beam at CERN) or protons (FASER2, which is expected to be placed near the ATLAS interaction point). Sterile states $N_{1,2}$ are produced mainly from the decays of the mesons produced in these collisions (see Sec. 3.2) and they travel a macroscopic distance until they reach the detector that can measure their decay products.

To estimate the regions in parameter space in which the dipole dominates production and/or decay, we will focus on $N_2$ (we will comment later on what happens for $N_1$) and proceed as follows: concerning production, we will consider the number of $N_2$ particles produced in mesons decays due to either the dipole ($N_d$) or the mixing ($N_\theta$): the dipole dominates in the region in which $N_d > N_\theta$. The computation of $N_d$ and $N_\theta$ can be done by taking the number of mesons produced per collision at the experiments considered (see [22] and App. D) and multiplying them by the meson branching ratio into $N_2$: see App. C.2 for the explicit decay widths of mesons into sterile states and App. C.3 for more details on the computation of the number of events. As for decays, we again focus on $N_2$ and consider the dipole to dominate when $\Gamma_d > \Gamma_\theta$, where $\Gamma_d \equiv \Gamma(N_2 \to N_1 \gamma) + \sum_\alpha \Gamma(N_2 \to \nu_\alpha \gamma)$, with the explicit expressions given in Eq. (35), while $\Gamma_\theta$ is the sum of the decay widths for all the channels due to the mixing only, whose explicit expressions can be found App. C.1, while more details on the behaviour of the sterile decays can be found in App. C.3.

---

[2]We note in passing that the dimension-five dipole operator couples sterile neutrinos to the $Z$ boson as well. However, collider precision measurements of the $Z$ properties do not set a significant bound on $d$ [52], therefore in this paper we focus only on neutrino couplings to the photon.

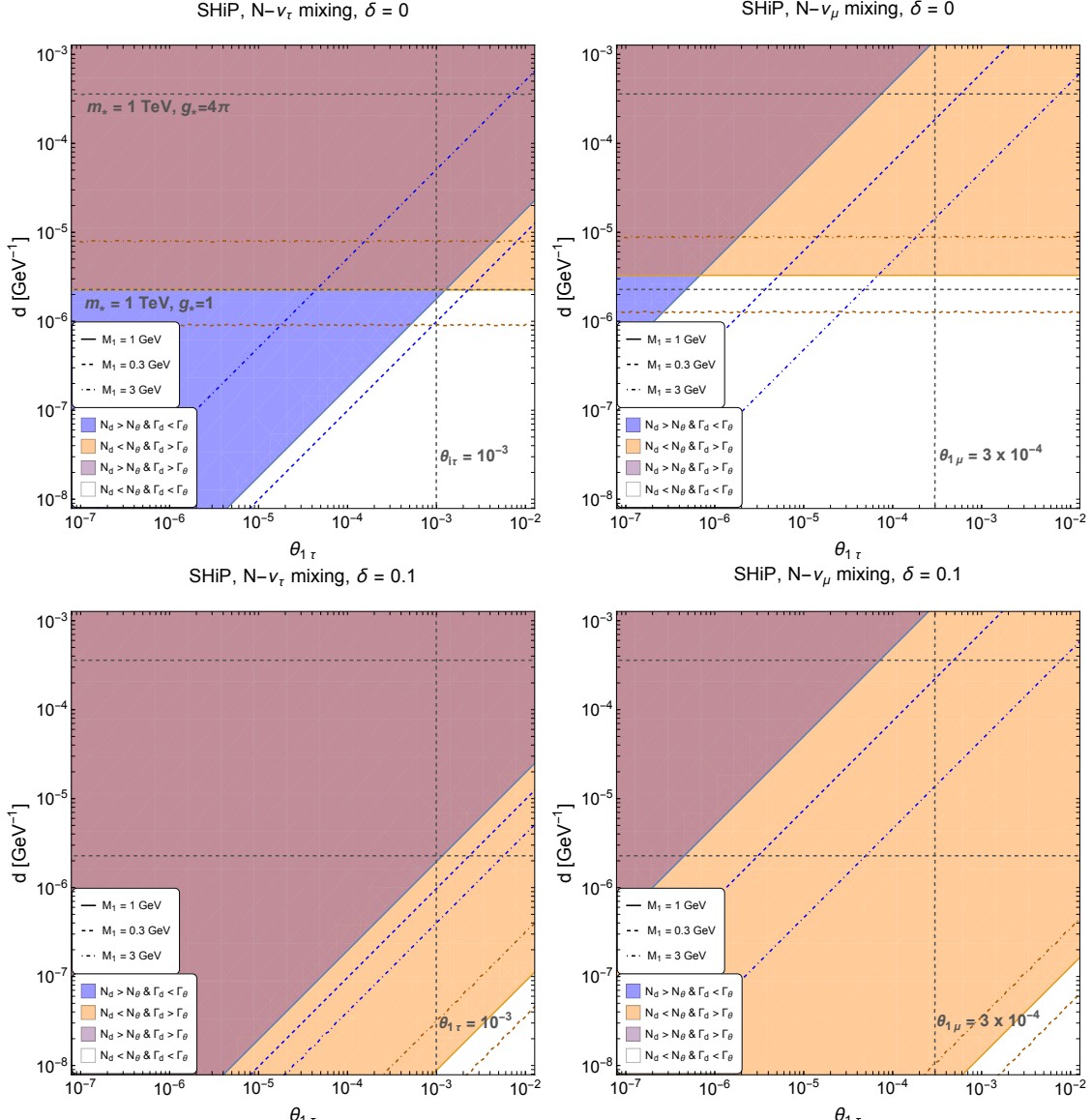

Figure 1: Regions in which $N_2$ production is dominated by the dipole, $N_d > N_\theta$ (above the blue lines), and where $N_2$ decays are dominated by the dipole, $\Gamma_d > \Gamma_\theta$ (above the orange lines), for three different values of $M_1$. The light blue/orange colour shading corresponds to the case $M_1 = 1$ GeV. In the wine (white) region, the dipole (the mixing) dominates both $N_2$ production and decays. We focused on the SHiP case, fixing a mass splitting $\delta = 0$ (0.1) in the upper (lower) panels, and non-zero mixing with a single flavour, $\theta_{i\tau} \neq 0$ ($\theta_{i\mu} \neq 0$) in the left (right) panels. The lower (upper) horizontal dashed gray line corresponds to a dipole generate by new physics at scale $m_* = 1$ TeV in a weak (strong) coupling regime, $g_* = 1$ ($g_* = 4\pi$). The vertical dashed gray line is the maximal allowed mixing for $M_1 = 1$ GeV (in the mixing-only scenario).

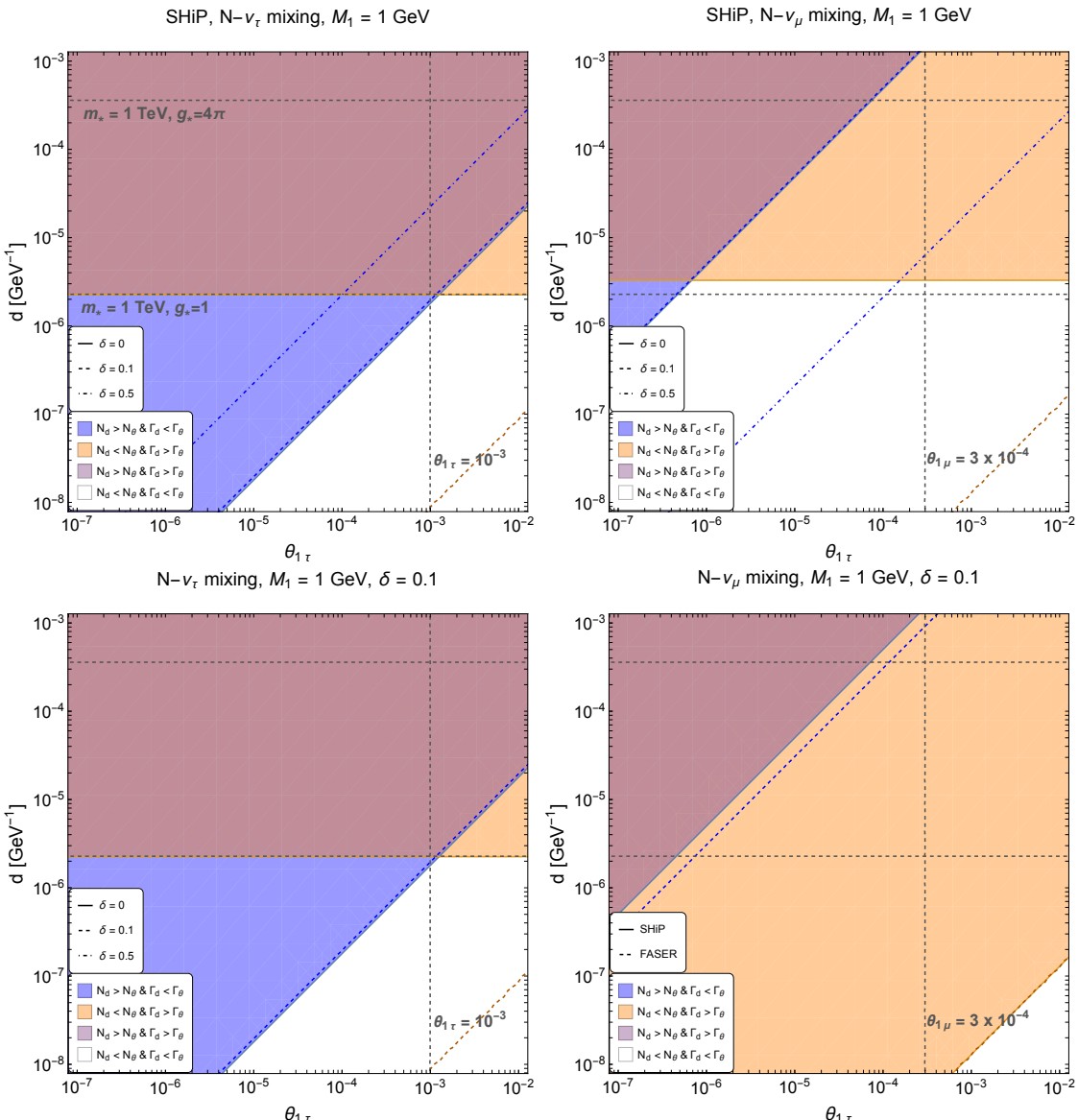

Figure 2: Same as in Fig. 1, but for different choices of parameters. In the upper panels, we focus on the SHiP experiment and fix $M_1 = 1$ GeV, while considering three different values of $\delta$ (the shading corresponds to $\delta = 0$). In the lower panels, we fix $M_1 = 1$ GeV and $\delta = 0.1$, and compare the SHiP and FASER2 experiments (the shading corresponds to SHiP). The curves for NA62 (not shown) basically coincide with those for SHiP.

Our results are shown in Figs. 1 and 2 for different choices of parameters. In both figures, we have $N_d > N_\theta$ above the blue lines and $\Gamma_d > \Gamma_\theta$ above the orange lines. In the wine-shaded region (where blue and orange shadings superimpose) the dipole dominates both production and decays. The opposite situation occurs in the white region, where the mixing dominates both production and decays. We also have blue-shaded regions, in which the dipole dominates production but not decays, and orange-shaded regions, in which the dipole dominates decays but not production. On the left (right) panels we turn on the mixing with the $\tau$ ($\mu$) flavour only. We have explicitly checked that for mixing with the $e$ flavour the results are essentially the same as for the $\mu$ flavour, so we do not show them. The two horizontal dashed gray lines show two representative values for the dipole coefficient, obtained using the estimate in Eq. (2) and fixing $m_\star = 1$ TeV, with $g_\star = 1$ for a weakly coupled UV completion, and $g_\star = 4\pi$ for a strongly coupled UV completion. The vertical dashed gray line shows, instead, the approximate upper limit on the mixing angles: $\theta_{1\tau} \simeq 10^{-3}$ and $\theta_{1\mu} \simeq 3 \times 10^{-4}$. These are taken from [2] and are the maximum allowed mixing for the representative value $M_1 = 1$ GeV. We stress once more that, strictly speaking, these limits have been computed assuming that only the mixing is present and so they are valid only in the white regions. In the coloured regions, a dedicated analysis would be necessary to determine the actual constraints in the presence of both dipole and mixing: as already mentioned, such analysis goes beyond the scope of the paper.

In Fig. 1 we fix $\delta = 0$ (upper panels) and $\delta = 0.1$ (lower panels), use Eq. (32) to compute $\theta_{2\alpha}$ as a function of $\theta_{1\alpha}$, and consider what happens at the SHiP experiment for three different values of the lightest sterile mass: $M_1 = 1$ GeV (solid lines, for which we colour-shade the regions in which $N_d > N_\theta$ and/or $\Gamma_d > \Gamma_\theta$), $M_1 = 0.3$ GeV (dashed lines) and $M_1 = 3$ GeV (dot-dashed lines). The region in which the dipole dominates both production and decays depends crucially on the choice of parameters. Concerning production (blue), we see that the ordering of the continuous, dashed and dot-dashed lines is not always monotonous (i.e. there are cases in which the $M_1 = 0.3$ GeV and 3 GeV lines lie on different sides of the $M_1 = 1$ GeV line, while in other cases they lie on the same side). This is due to the fact that the ratio $N_d/N_\theta$ is not a monotonic function of $M_1$, because the dipole and mixing production channels have different thresholds and different mass dependence, see App. C.3. For decays, the ratio $\Gamma_d/\Gamma_\theta$ is a monotonic function of $M_1$, and the ordering of the lines is always the same in the different panels. We also notice that, when $\delta = 0$, the orange lines are horizontal, while for $\delta \neq 0$ they are oblique. This is due to the fact that, in the former case, the decay channel $N_2 \to N_1 \gamma$ is kinematically closed, leaving only $N_2 \to \nu\gamma$ open. This is proportional to $|\theta_{1\alpha}|^2$,[3] so the dependence on the mixing cancels when comparing with $\Gamma_\theta$, which is also proportional to $|\theta_{1\alpha}|^2$. This behaviour (horizontal orange lines) is the one we would observe in the analogous plots for the $N_1$ particle, since in this case the only open decay channel is $N_1 \to \nu\gamma$: for this reason we decided to rather display plots for the $N_2$ production and decays.

Turning to Fig. 2, in the upper panels we show a different slice of parameter space, in which we fix $M_1 = 1$ GeV and let $\delta$ take the values 0, 0.1 and 0.5. We again see the appearance of horizontal orange lines for $\delta = 0$, while the dot-dashed lines corresponding to $\delta = 0.5$ are not visible in the plot, as they lie in the bottom-right corner. The experiment considered is again SHiP. In the lower panel we instead analyse what happens at different experiments, fixing $M_1 = 1$ GeV and $\delta = 0.1$. The experiments considered are FASER2 and SHiP. Only production changes between the two experiments, simply because the decay widths do not depend on the experiment considered. We do not show the curve for NA62 because, like SHiP, it consists on a 400 GeV proton beam scattering against a target and, despite the target being different in the two experiments, the overall number of $N_2$ particles produced is very similar.

---

[3]Although we are considering $N_2$, one can use Eq. (32) to express $\theta_{2\alpha}$ as a function of $\theta_{1\alpha}$.

The main conclusion of our analysis is that, although it is possible to have the dipole to dominate both production and decays, the extension of the region in which this happens depends strongly on the parameters $M_1$ and $\delta$ (or equivalently $M_2$). Clearly, there is always a limit $\theta_{i\alpha} \to 0$ in which the dipole dominates: this is the case considered in Refs. [22,52]. In the opposite limit, $d \to 0$, one is left with the mixing and the EW dipole, but the latter is typically subdominant because it is loop suppressed. In particular, all the bounds discussed in Sec. 2.4 are expected to apply, as they are derived assuming mixing only.

## 3.2 Dipole signal at intensity frontier experiments

We now turn to the study of the limits and of the sensitivity on the parameter space due to past and future experiments at the intensity frontier. Since we focus on sterile neutrinos with mass below a few GeV, we will consider only $N_{1,2}$ production via mesons decays: possible contributions from parton interactions have been shown to be subdominant at fixed target experiments [53], and are expected to be subdominant also at the LHC [54]. Since we are interested in probing the dipole operator, we will consider as signal the production of a photon inside the detector. Clearly, this is meaningful only provided the experiment considered will be able to (i) detect photons and (ii) distinguish the photon from the background. The last condition is typically satisfied provided the photon energy is larger than an experimental threshold $E_\gamma$, that varies according to the experiment. We will denote $P_{E_\gamma}^X$ the probability to pass this requirement, with $X = N_1, N_2$ the particle that produces the photon via its decay.

The computation of the number of events is more complicated with respect to the case considered in Refs. [22,55], which focused on the zero-mixing limit, $\theta_{i\alpha} = 0$. This is because $\theta_{i\alpha} \neq 0$ induce additional production/decay modes for the sterile neutrinos, including the case in which $N_1$ particles are produced by the dipole via $N_2 \to N_1 \gamma$, and then decay via $N_1 \to \nu_\alpha \gamma$, which depends on both the dipole and the mixing. Note also that, when the sterile mass splitting vanishes, $\delta = 0$, then the channel $N_2 \to N_1 \gamma$ is kinematically close, and non-zero mixing is necessary to produce a signal.

We start by introducing the probability for a particle $X$, produced at position $x$, to decay between some $y \geq x$ and $y + dy$:

$$p_X(x,y)dy = \frac{e^{-(y-x)/L_X}}{L_X}dy\,, \qquad (36)$$

where $L_X = \beta c \gamma \tau_X$ is the decay length of the particle $X$ in the lab frame, with $\tau_X$ the lifetime in the particle rest frame, $\gamma$ the boost factor from the particle rest frame to the lab frame, and $c\beta$ the particle velocity in the lab frame.

In a generic point of parameter space, we can distinguish between five "populations" of events, classified according to the decay that generates the signal and where the mother particle has been produced:

1. $N_2$ is produced via mesons decays at the interaction point and decays via $N_2 \to \nu_\alpha \gamma$ inside the detector. The number of events is

$$N_{\text{events}}^{(1)} = \sum_M \mathcal{N}_M \, \text{BR}(M \to N_2) \langle P_{N_2\,\text{inside}} \rangle\,, \qquad (37)$$

where $\mathcal{N}_M$ is the number of mesons of type $M$ produced at the experiment considered, $\text{BR}(M \to N_2)$ the meson branching ratio into $N_2$, and $P_{N_2\,\text{inside}}$ the probability for the $N_2 \to \nu_\alpha \gamma$ decay to happen inside the detector, with a photon above threshold,

$$P_{N_2\,\text{inside}} = \left(e^{-L_{in}/L_{N_2}} - e^{-L_{out}/L_{N_2}}\right) P_{E_\gamma}^{N_2} \sum_\alpha \text{BR}(N_2 \to \nu_\alpha \gamma)\,. \qquad (38)$$

In this equation, the term inside brackets is the total probability for $N_2$ to decay inside the detector, i.e. $\int_{L_{in}}^{L_{out}} dx\, p_{N_2}(0,x)$, where $L_{in}$ and $L_{out}$ are the distances from the interaction point at which $N_2$ enters and exits the detector, respectively. The symbol $\langle \cdot \rangle$ in Eq. (37) denotes the average over all simulated momenta, necessary because the quantities $L_{N_2}$ and $P_{E_\gamma}^{N_2}$ depend on the momentum of the $N_2$ particle considered.

2. $N_1$ is produced via mesons decays at the interaction point and decays via $N_1 \to \nu_\alpha \gamma$ inside the detector. The number of events is

$$N_{\text{events}}^{(2)} = \sum_M \mathcal{N}_M \, \text{BR}(M \to N_1) \langle P_{N_1 \text{ inside}} \rangle \,, \tag{39}$$

and can be obtained from Eqs. (37)-(38), simply substituting $N_2 \to N_1$.

3. $N_2$ is produced via mesons decays at the interaction point and decays via $N_2 \to N_1 \gamma$. We now have three possible sources of signal:

   (a) $N_2 \to N_1 \gamma$ happens before the detector while $N_1 \to \nu_\alpha \gamma$ happens inside the detector. The total number of events is given by

$$N_{\text{events}}^{(3)} = \sum_M \mathcal{N}_M \, \text{BR}(M \to N_2) \langle P_{N_2 \text{ before},N_1 \text{ inside}} \rangle \,, \tag{40}$$

   where the probability is

$$\begin{aligned} P_{N_2 \text{ before},N_1 \text{ inside}} = \int_0^{L_{in}} dx\, p_{N_2}(0,x) \int_{L_{in}}^{L_{out}} dy\, p_{N_1}(x,y) P_{E_\gamma}^{N_1} \\ \times \text{BR}(N_2 \to N_1 \gamma) \sum_\alpha \text{BR}(N_1 \to \nu_\alpha \gamma) \,. \end{aligned} \tag{41}$$

   The average $\langle \cdot \rangle$ is now taken over all $N_1$ and $N_2$ simulated momenta.

   (b) $N_2 \to N_1 \gamma$ happens inside the detector while $N_1 \to \nu_\alpha \gamma$ happens outside the detector. The total number of events is

$$N_{\text{events}}^{(4)} = \sum_M \mathcal{N}_M \, \text{BR}(M \to N_2) \langle P_{N_2 \text{ inside},N_1 \text{ outside}} \rangle \,, \tag{42}$$

   where the probability is

$$P_{N_2 \text{ inside},N_1 \text{ outside}} = \int_{L_{in}}^{L_{out}} dx\, p_{N_2}(0,x) \int_{L_{out}}^{\infty} dy\, p_{N_1}(x,y) P_{E_\gamma}^{N_2} \text{BR}(N_2 \to N_1 \gamma) \,. \tag{43}$$

   (c) Both $N_2 \to N_1 \gamma$ and $N_1 \to \nu_\alpha \gamma$ happen inside the detector. The number of events is

$$N_{\text{events}}^{(5)} = \sum_M \mathcal{N}_M \, \text{BR}(M \to N_2) 2 \langle P_{N_2 \text{ inside},N_1 \text{ inside}} \rangle \,, \tag{44}$$

   where the factor of 2 accounts for the two photons produced in each such event, and the decay probability is

$$\begin{aligned} P_{N_2 \text{ inside},N_1 \text{ inside}} = \int_{L_{in}}^{L_{out}} dx\, p_{N_2}(0,x) \int_{L_{in}}^{L_{out}} dy\, p_{N_1}(x,y) P_{E_\gamma}^{N_2} P_{E_\gamma}^{N_1} \\ \times \text{BR}(N_2 \to N_1 \gamma) \sum_\alpha \text{BR}(N_1 \to \nu_\alpha \gamma) \,. \end{aligned} \tag{45}$$

We will consider in our analysis the past beam-dump experiments BEBC [6] and CHARM-II [7], and the already-mentioned future experiments NA62 (in beam-dump mode), SHiP and FASER2. Details on the experiments geometry, meson multiplicity and analysis follow those used in [22] and can be found in App. D. We do not show the region excluded by NuCal (considered in [22]) because it is typically subdominant with respect to the other exclusions. We also do not show the bound coming from BaBar, which typically excludes $d \gtrsim 8 \times 10^{-4}$ GeV$^{-1}$. For consistency of the effective field theory, in our plots we will focus on the region with a dipole Wilson coefficient $d < 10^{-3}$ GeV$^{-1}$, without showing the BaBar limit.

For simplicity, in our simulations we will consider production of sterile neutrinos only from the two-body mesons decays $V^0 \to N_1 N_2$, $V^0 \to N_i \nu_\alpha$, $P^0 \to N_i \nu_\alpha$ and $P^\pm \to N_i \ell^\pm$, where $V$ and $P$ denote a vector and pseudoscalar meson, respectively, while $\ell$ is a charged lepton. Additional three-body decays have been showed to give rather small contributions [22, 53] and will not be considered. We include $V^0 = \{\rho, \omega, \phi, J/\psi, \Upsilon\}$, $P^0 = \{\pi^0, \eta, \eta'\}$, and $P^\pm = \{D^\pm, D_s^\pm, B^\pm\}$. The decay widths that enter in the computation of the number of events are those in Eq. (35) and those listed in Apps. C.1-C.2.

Our final results are presented in Figs. 3-5, in which we show the present bounds (shaded regions) and future sensitivities to the photon signal (solid lines) in the $(M_1, d)$ plane, for various values of the sterile mass splitting $\delta$ and of the mixing angles $\theta_{1\alpha}$. The lines shown correspond to a sensitivity computed at 95% CL. For future experiments, this corresponds to the 3-event line (since the assumption is of no background). For the past experiments, they correspond to a 3-event (BEBC) and 145-event (CHARM-II) line. These numbers have been computed using the prescription in App. B of [23], using as an input the estimations of the background from the two collaborations.

In the cases with a non-zero mixing, one observes that some sensitivity regions extend vertically down to arbitrarily small values of $d$, for a certain mass window around $M_1 \simeq 1$ GeV. This behaviour can be easily understood: photon production is controlled by $d^{em} = d^{NP} + d^{EW}$, as shown in Eq. (35). Therefore, even when $d^{NP} \to 0$, the EW contribution can be enough to produce a detectable number of events. This is precisely what happens in the vertical regions, in which the dependence on $d$ disappears, and the photons in the detector are produced solely by the EW contribution.

In the upper panel of Fig. 3 we show the case of vanishing mixing, $\theta_{i\alpha} = 0$, fixing $\delta = 0.1$. For this choice, production happens via the decays of neutral vector mesons, $V^0 \to N_1 N_2$, and the signal is generated by the channel $N_2 \to N_1 \gamma$. We observe that, for our choice of $\delta$, CHARM-II and BEBC do not exclude any region of the parameter space shown.[4] One can see that NA62 and FASER2 will be sensitive to the region between the two target values of the dipole coefficient, corresponding to $m_* = 1$ TeV and either strong or weak coupling, $g_* = 4\pi$ or $g_* = 1$, as long as $M_1 \lesssim 0.3$ GeV. On the other hand, SHiP will be able to explore significantly smaller values for the dipole, down to $d \gtrsim 2 \times 10^{-7}$ GeV$^{-1}$, for a sterile mass as large as $M_1 \sim 1$ GeV.

---

[4]Comparing with the upper left panel of Fig. 1 of [22], this may come as a surprise, since in that reference there is a region excluded by CHARM-II and BEBC. The reason for the difference stems from the different definitions of the parameter $\delta$. Our $\delta = 0.1$ corresponds to $\delta \simeq 0.23$ of Ref. [22], a value sufficient to increase our decay width by roughly a factor of 10, with respect to the upper left panel of Fig. 1 of the reference, justifying the different behaviour of the excluded regions. This also explains the slightly different shape of the sensitivities shown in Fig. 3 with respect to the corresponding sensitivities in Fig. 1 of [22].

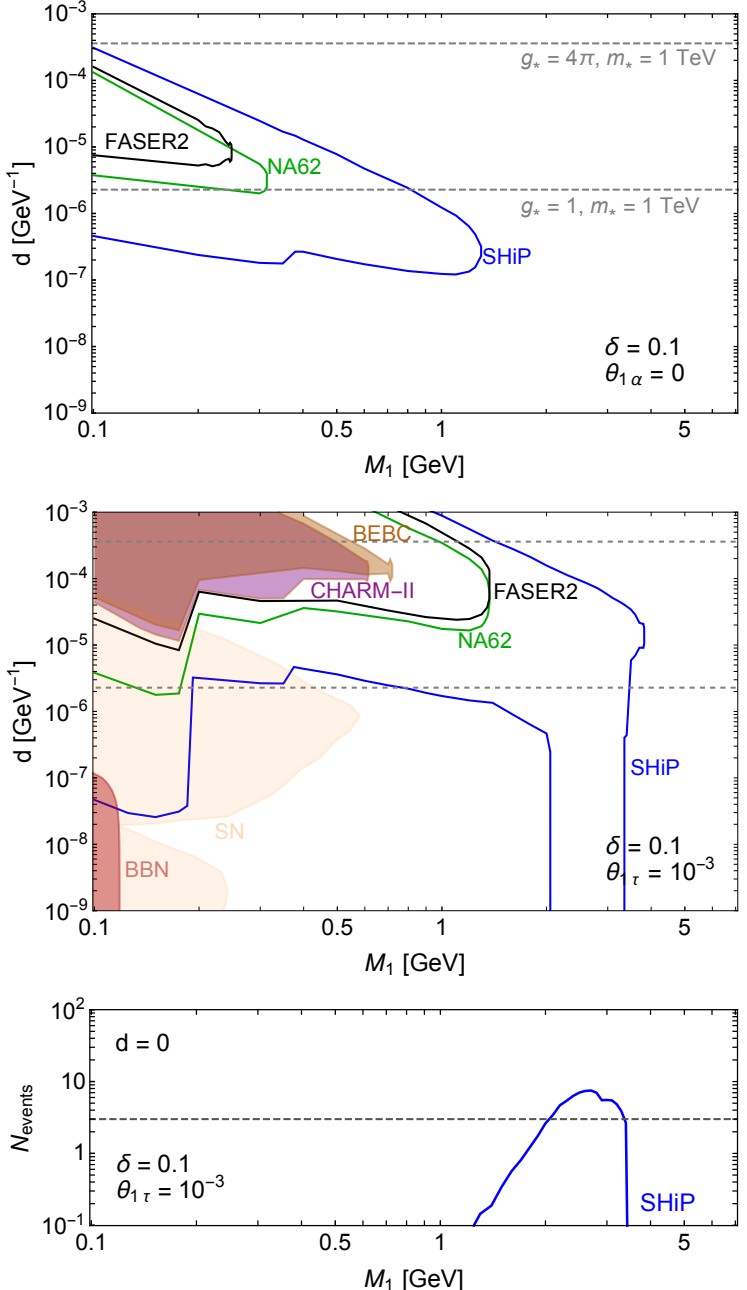

Figure 3: Sensitivity to the photon signal of the future experiments FASER2, NA62 and SHiP (coloured lines) and regions excluded by the past experiments CHARM-II and BEBC (coloured regions), fixing $\delta = 0.1$ and either no mixing (upper panel) or mixing with the $\tau$ flavour (middle panel). We also show the regions excluded by limits from supernovæ (SN) and Big Bang Nucleosynthesis (BBN). The lower (upper) horizontal dashed gray line corresponds to a dipole generate by new physics at scale $m_* = 1$ TeV in a weak (strong) coupling regime, $g_* = 1$ ($g_* = 4\pi$). The lower panel shows the number of events generated by the EW dipole (limit $d \to 0$), for the same parameters as in the middle panel. When $N_{events} > 3$ (dashed horizontal line), the corresponding sensitivity extends to arbitrarily small values of $d$ (vertical region in the middle panel).

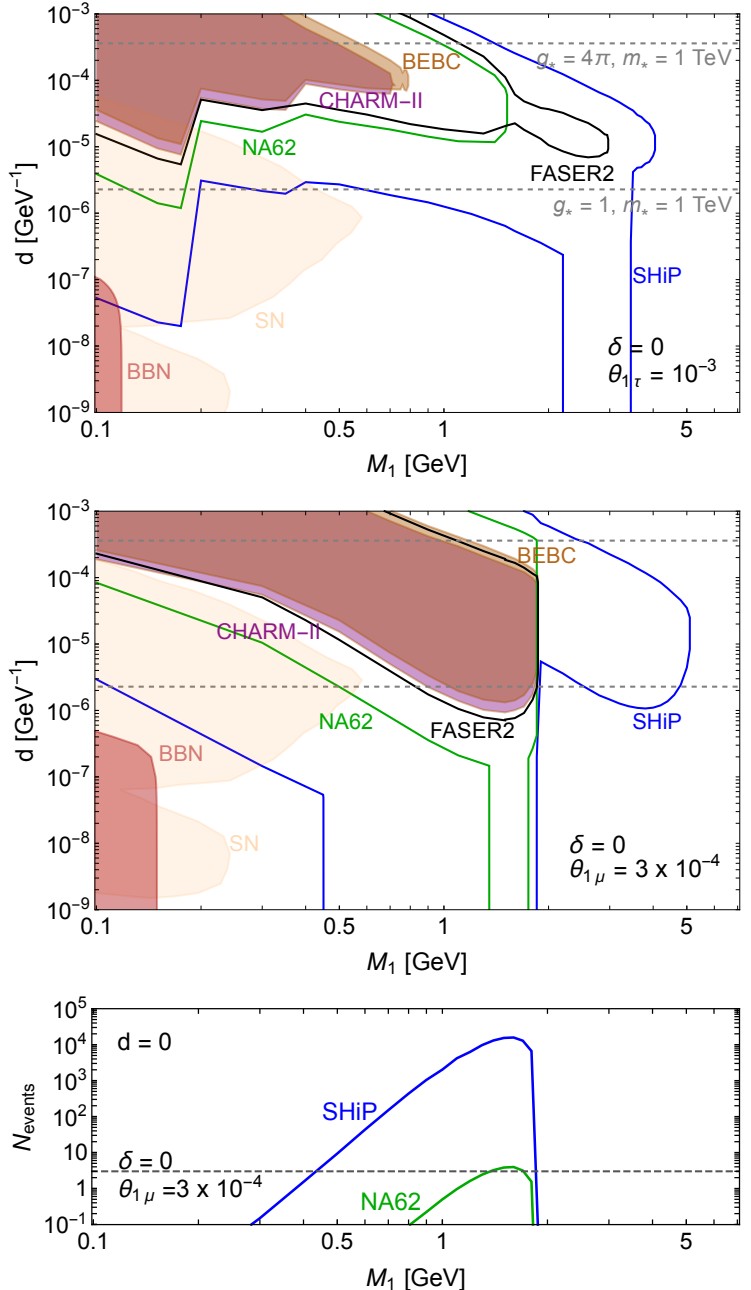

Figure 4: Upper panel: Same as in the middle panel of Fig. 3, but choosing $\delta = 0$ (instead of $\delta = 0.1$). Middle panel: same as the upper panel, but mixing with the $\mu$ flavour (instead of the $\tau$ flavour). Lower panel: number of events generated by the EW dipole, for the same parameters as in the middle panel, but in the limit $d \to 0$.

Moving to the middle panel of Fig. 3, we keep $\delta = 0.1$ but turn on a mixing with the $\tau$ flavour, fixed to the maximum allowed value for a reference mass $M_1 = 1$ GeV, that is $\theta_{1\tau} = 10^{-3}$. This adds new mixing-induced production and decay channels to those already considered in the upper panel. The shape of the sensitivity regions are completely different with respect to the previous case and, in particular, there are now regions excluded by CHARM-II and BEBC. A common feature of all the curves is the "jump" in sensitivity appearing at $M_1 \simeq 0.2$ GeV, where the channel $D_s^{\pm} \to N_i \tau^{\pm}$ closes. We can also see the effect of smaller thresholds: for instance, the small feature around $M_1 \simeq 0.4$ GeV is due to the dipole-induced decays $\rho \to N_1 N_2$ and $\omega \to N_1 N_2$ closing. The shape and extension of the curves depend strongly on the experiment via its geometry and the number of sterile produced. In particular, these affect the value of $M_1$ at which the experimental sensitivity ceases to be effective, the largest being $M_1 \sim 3{-}4$ GeV for SHiP. In this panel we also see a region excluded by supernovæ (SN) (light orange) and by Big Bang Nucleosynthesis (BBN), which will be discussed in detail in Sec. 3.3. These limits do not appear in the upper panel of Fig. 3: concerning BBN, the excluded region lies below the range for $d$ shown in the plot; concerning SN, to the best of our knowledge, the limit has not been computed in this case (i.e. for sterile neutrinos coupled only to the photon, without active-sterile mixing).

As already mentioned, in the case of SHiP, the vertical region that extends to arbitrarily small values of $d$ is due to the photon signal produced by the EW contribution to the dipole. To illustrate this interpretation, in the lower panel of Fig. 3 we show the number of events as a function of $M_1$, for the same parameters as in the middle panel, but fixing $d = 0$. The interval in $M_1$ where we have more than three events precisely matches the vertical region in the middle panel. For the remaining experiments, the EW contribution is too small to give a measurable signal.

Coming to Fig. 4, the upper panel differs from the middle panel of Fig. 3 only for the value of $\delta$: we can see that only marginal differences appear, even though in the $\delta = 0.1$ case there is the additional decay channel $N_2 \to N_1 \gamma$, that may generate photons in the detector and $N_1$ particles that can themselves give a signal. In the terminology used above, the signal for $\delta = 0$ is generated by populations 1 and 2, while the signal for $\delta = 0.1$ is generated by all 5 populations. Nevertheless, at least for $\delta = 0.1$ we see that the effect of the populations 3–5 is small in most of the $M_1 - d$ plane. Indeed, as the mass splitting becomes smaller, the channel $N_2 \to N_1 \gamma$ progressively closes, and one must recover the $\delta = 0$ result continuously. Also in Fig. 4 we show the SN and BBN constraints, to be discussed later in Sec. 3.3.

In the middle panel of Fig. 4, we consider a mixing with the $\mu$ flavour, in contrast with the $\tau$ flavour in the upper panel. One main difference between the two panels stems from the different position of the meson thresholds. For the $\mu$ flavour, the channels $D^{\pm} \to N_i \mu^{\pm}$ and $D_s^{\pm} \to N_i \mu^{\pm}$ remain open all the way up to $M_1 \simeq 2$ GeV. For the $\tau$ flavour, instead, the analogous threshold appears at $M_1 \simeq 0.2$ GeV, due to the $D_s^{\pm}$ decay (the $D^{\pm}$ decay closes for $M_1 \lesssim 0.1$ GeV and does not appear in our plot). For mixing with the $\mu$ flavour, only the SHiP experiment has a sensitivity that extends beyond the $D/D_s$ threshold.

A second important difference is that photon events from the EW dipole are much more frequent in the case of $\mu$ flavour, and they peak at lower values of $M_1$. This is illustrated in the lower panel of Fig. 4, where we show the number of events obtained when the NP dipole $d$ is switched off, for SHiP and NA62. While the latter experiment is barely sensitive to the photons produced by the EW contribution, for SHiP we expect up to $\mathcal{O}(10^4)$ events for $M_1 \sim 2$ GeV.

We now turn to Fig. 5, where we present the sensitivity of the SHiP experiment fixing a mixing with the $\mu$ flavour and varying $\delta$ (upper panel), or fixing $\delta = 0$ and varying the mixing angles (lower panel). In the upper panel, the curves for $\delta = 0$ (continuous) and $\delta = 0.1$ (dot-dashed) are pretty similar, a behaviour that was already seen for the $\tau$ mixing case (compare

the lower panel of Fig. 3 and the upper panel of 4). The situation is significantly different for $\delta = 0.9$ (dashed line). This can be understood by observing that, for $\delta = 0.9$, we have $M_2 = M_1(1 + \delta)/(1 - \delta) = 19\,M_1$. Therefore, as $M_1$ grows, the heaviest sterile neutrino soon becomes too heavy to be produced in meson decays. Since the signal is now produced by $N_1$ in most of the parameter space, the total number of events is roughly half the one we have for smaller $\delta$, and the sensitivity curves shift to larger values of $d$. In the same plot we show the SN bound, which has been derived only for a single sterile neutrino (of mass $M_1$), and the BBN bound, which depends on $\delta$. For this reason, we display three shaded regions for BBN, with a solid contour for $\delta = 0$, a dot-dashed contour for $\delta = 0.1$, and a dashed contour for $\delta = 0.9$.

In the lower panel of Fig. 5, we consider both mixing with individual flavours and, motivated by our discussion on oscillation data in Sec. 2.4, cases in which $\theta_{i\mu} = \theta_{i\tau}$ (dubbed $\mu + \tau$ in what follows), or $\theta_{ie} = \theta_{i\mu} = \theta_{i\tau}$ ($e + \mu + \tau$). In such multi-flavour cases, the mixing is fixed to the maximal allowed value for the most constrained flavour. Whenever the mixing includes the $e$ or $\mu$ flavours, the sensitivities are pretty similar. This is because both the electron and the muon are sufficiently light that their mass difference does not play a crucial role in determining the sensitivity curve, except for a $\sim 100$ MeV difference between the $e$ and $\mu$ curves at the $D/D_s$ threshold, close to the end of the vertical region, around $M_1 \simeq 2$ GeV, due precisely to the $e - \mu$ mass difference (difference which is inherited by the $e + \mu + \tau$ and $\mu + \tau$ curves as well). The sensitivity curve for mixing with the $\tau$ flavour instead has thresholds appearing for significantly different $M_1$, therefore it results to be very different with respect to the $e$ and $\mu$ curves. Comparing the $\mu$ case (blue) with the $\mu + \tau$ curve (purple), the more pronounced differences arise in the regions $M_1 \lesssim 0.2$ GeV and $2$ GeV $\lesssim M_1 \lesssim 3.5$ GeV, precisely where the sensitivity of the $\tau$ line (red) is many orders of magnitude better than the sensitivity of the blue line. Nevertheless, even in these regions, the $\mu + \tau$ line does not reach the sensitivity of the $\tau$ line, because the maximal allowed mixing angles are smaller. The regions excluded by SN (BBN) are shown using the universal mixing case (mixing with electron only case): these exclusion regions would shift for the other mixing cases.

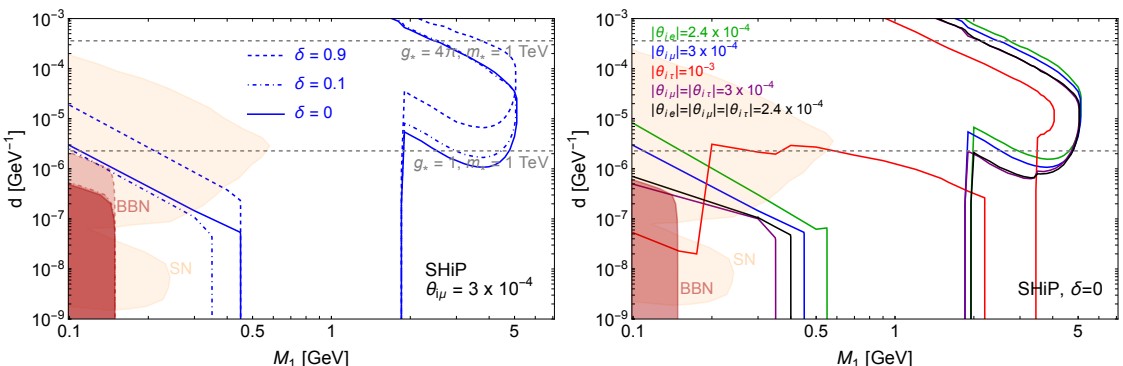

Figure 5: Sensitivity of the SHiP experiment to the dipole coefficient, as a function of the lightest sterile mass. In the upper panel, we consider a mixing with the $\mu$ flavour only, and vary the mass splitting $\delta$. In the lower panel, we set $\delta$ to zero, and consider a mixing with various combinations of lepton flavours. For definiteness, the SN excluded region corresponds to the universal mixing case (black) and the BBN excluded region to the electron mixing (green).

## 3.3 Complementary constraints on the dipole portal

In addition to the limits coming from flavour observables (see Sec. 2.4) and those from CHARM-II and BEBC discussed in the previous section, the parameter space can also be bounded by looking at two other types of phenomena:

(i) the active-sterile dipole generated by the mixing, see Eq. (13), allows for neutrino beams to be upscattered into sterile neutrinos, in such a way that limits can arise from neutrino experiments, if the detector is able to measure the photons coming from the decay $N_i \to \nu_\alpha \gamma$;

(ii) cosmological (BBN) and astrophysical (supernovæ) data can bound the sterile neutrino total lifetime: in the former case, because sterile neutrino decays into SM states can alter the $^4$He and deuterium yields; in the latter case, because the sterile neutrino production and the following energy drain can alter the cooling rate of a supernova.

These limits have been comprehensively computed in Ref. [23] (and later refined in [24–42], see also [22,55]) assuming directly a low energy active-sterile dipole operator, $d_\alpha N \sigma^{\mu\nu} \nu_\alpha F_{\mu\nu}$ with $F_{\mu\nu}$ the photon field strength. This may arise from a dim-six operator $N \sigma^{\mu\nu}(L_\alpha H)B_{\mu\nu}$ which is present independently from the active-sterile mixing. This is in contrast with our scenario, where both the dim-five sterile-sterile operator and the EW loops induce an active-sterile dipole proportional to the active-sterile mixing. Since these limits turn out to be relevant only for very large $d$, or for small $M_1 \simeq$ a few 100 MeV, we will neglect $d^{EW}$ in the following.

Borrowing from Ref. [22], the active-sterile dipole coefficients are called $d_\alpha$, and translate to the notation used in the present paper as $d_\alpha = d\,\theta_{i\alpha}/2$, i.e. in our case the active-sterile dipole is mixing-suppressed. In the case of upscattering, this means that the amplitude of the process is proportional to $|d|^2|\theta_{i\alpha}|^2$ (with one factor arising from the photon exchange with the target necessary to convert the incoming active neutrino into a sterile neutrino, and the other arising from the $N_i \to \nu_\alpha \gamma$ decay). As a consequence, the upscattering process is too suppressed to give any relevant limit. More concretely, upscattering would limit the region $d \gtrsim 10^{-3}$ GeV$^{-1}$, not shown in Figs. 3–5.

On the contrary, astrophysical and cosmological limits arising from supernovæ and BBN may be complementary to accelerator experiments, since they typically exclude regions with smaller dipole/mixing couplings. In the case of supernovæ (SN), the bound arises from the study of the cooling efficiency [23,37]. In the limit of very small active-sterile dipole coupling, just a few sterile neutrinos are produced, so the cooling is inefficient and no bound arises. For very large coupling, sterile neutrinos may be trapped inside the SN and be reconverted into active neutrinos, with no impact on cooling and hence, again, no bound. The region between these two limiting cases is the one in which the SN bound is relevant, at least for sterile masses sufficiently small to be produced inside the SN ($M_i \lesssim 0.6$ GeV). We show these limits with a light shaded orange colour in Figs. 3–5. The upper "lobe" of the bound is derived considering cooling of SN of type IIP [39], while the lower "lobe" is derived using SN1987A [37]. In both cases, the limit has been computed considering a single sterile neutrino with $d_e = d_\mu = d_\tau$, and we replaced the single mixing declared in each figure into these three coefficients. We stress once more that the limits shown have been computed in [37, 39] considering only a direct active-sterile dipole and hence, strictly speaking, cannot be applied in our case, in which both sterile-sterile dipole and active-sterile mixing are present. Given the complexity of the computation, however, we will stick to this simple estimate.

Let us conclude this section discussing the limits coming from BBN. We follow the strategy of Ref. [23] and require that $\max(\tau_1, \tau_2) < 1$ s, where $\tau_i$ is the total lifetime of $N_i$. This bound guarantees that, even if the sterile neutrinos thermalise in the primordial plasma, they decay before the onset of BBN and do not spoil experimental predictions on the yields of primordial

light nuclei. The regions excluded by this bound are marked by the label BBN in Figs. 3–5, and they are limited to small dipole coefficients and masses $M_1 \lesssim 0.15$ GeV.

# 4 Conclusions

May photons provide an access to light new physics, below the collider scale? The charge of new light particles is strongly constrained. On the other hand, even if neutral, such particles may interact with photons via higher-dimensional operators, or via quantum corrections. In the case of sterile neutrinos $N_i$, significant dipole interactions can be induced by a dimension-five operator, as well as by EW loops. They provide an alternative portal to produce and detect sterile neutrinos, beside the traditional portal offered by the mixing with SM active neutrinos. In this article we investigated the interplay between these two possibilities to discover sterile neutrinos, focusing on the mass range $M_i \sim 0.1 - 10$ GeV. In this window, mesons produced in proton collisions can copiously decay into $N_i$, which are typically long-lived and can subsequently decay in far detectors. We focused on dipole-driven decays, into a photon plus a lighter neutrino, which could be efficiently detected in present and near future experiments.

We began (section 2) by estimating the size of the sterile neutrino dipole, and by showing that its scale is naturally independent from the sterile mass scale. We carefully computed the dipole couplings of the various neutrino mass eigenstates, induced by the mass mixing between active and sterile states, taking into account direct and indirect bounds on the mixing angles, for the various lepton flavours.

We then identified (section 3.1 and appendices) the region of parameters where $N_i$ production is dominated by the dipole or, alternatively, by the mixing with SM neutrinos of each given flavour. An analogous analysis was conducted on the various $N_i$ decay channels induced by the dipole and by the mixing. In particular, we showed that there are regions where the dipole dominates both production and decay, even for mixing angles as large as currently allowed.

The experimental sensitivities to the dipole were presented in section 3.2. In the zero-mixing limit, the current NA62 data-taking and the future FASER2 experiments have similar sensitivity, covering the region with sterile masses $M_i \lesssim 0.3$ GeV and dim-five dipole coefficient $d \gtrsim 2 \times 10^{-6}$ GeV$^{-1}$, corresponding to new charged states close to the TeV scale. The SHiP experiments will improve the sensitivity to $M_i \lesssim 1$ GeV and $d \gtrsim 10^{-7}$ GeV$^{-1}$.

On the other hand, a non-zero mixing can greatly enhance $N_i$ production, and thus strongly increase the detector sensitivity, so that SHiP may reach sterile neutrinos as heavy as $\simeq 5$ GeV, and a dipole coefficient as small as $d \sim 10^{-8} - 10^{-9}$ GeV$^{-1}$, corresponding to new charged states as heavy as $\sim 10^3$ TeV. However, the mixing also induces an additional contribution $d^{EW}$ to the electromagnetic dipole, from $W$/charged lepton loops, that may be sufficient to generate a detectable photon signal in the window $0.3$ GeV $\lesssim M_i \lesssim 3$ GeV, even in the limit $d \rightarrow 0$. On the one hand, this effect may overwrite part of the signal from the dim-five dipole; on the other hand, it constitutes an irreducible signature of the active-sterile mixing.

In the absence of mixing, one needs a non-zero mass splitting $\delta$ between the sterile neutrinos, in order for the radiative decay of the heaviest into the lightest to happen. In the presence of mixing, such mass splitting indicates a violation of lepton number, and it should be naturally small. However, even in the natural, lepton-number conserving limit $\delta = 0$, where the two sterile neutrinos form a Dirac state, the mixing allows for the radiative decay of the heavy state into SM neutrinos. Actually, the sensitivity to the dipole coefficient turns out to be slightly better when the splitting is small, as both sterile states contribute to the signal.

We carefully compared the sensitivity for different lepton-flavour configurations. The sterile mixing with the $\tau$ neutrino is allowed to be larger than with the $\mu$ and $e$ flavours but, on the other hand, the thresholds for charged-current production and decay of $N_i$ are larger because

of the heavier $\tau$ mass. As a consequence, whether the sensitivity to $d$ is better in the $\tau$ case, or not, depends on the specific value of $M_i$. We also analysed the case of equal mixing with the different flavours, which is motivated by requiring a minimal two-sterile seesaw mechanism in order to accommodate neutrino oscillation data. If some experiment will catch a sterile neutrino by the dipole, it will be possible to correlate the signal with the various flavour constraints.

We collected (section 3.3) various complementary bounds on the dipole of sterile neutrinos. In the absence of mixing they are typically very weak. In the presence of mixing, they can exclude sterile neutrinos only if they are lighter than $\sim 0.1 - 0.2$ GeV, almost independently from the value of the dipole coefficient. Thus, a vast region of parameters is left open, where the photon signal could allow for a discovery of GeV-scale sterile neutrinos, in near future data.

In the future, it could be interesting to extend our search for the sterile neutrino dipole to other mass ranges, and/or to different experimental setups, such as neutrino detectors. Analyses exist assuming the active-sterile mixing only, or other sorts of higher-dimensional operators, see e.g. [56] for the T2K near detector, and [57] for the DUNE near detector (see [58] for considerations on photon detection at DUNE).

Before concluding let us observe that, in the limit of vanishing active-sterile mixing, $\theta_{i\alpha} = 0$, the lightest state $N_1$ is stable, thanks to an unbroken sterile parity, $Z_{2N} : N_i \rightarrow -N_i$. Therefore $N_1$ becomes a potential dark matter candidate, that scatters on ordinary matter inelastically, converting into $N_2$ via a photon exchange. The phenomenology of such an inelastic dark matter candidate with a dipole interaction was explored in [41, 59]. Our results show that there are large regions of parameter space where a dipole signal could be detected in the $\theta_{i\alpha} = 0$ case but not in the $\theta_{i\alpha} \neq 0$ case, and viceversa. Therefore, it may be possible to distinguish the inelastic dark matter scenario from the active-sterile mixing scenario, provided that a dipole detection happens in one of these regions.

Notice also that our results strictly apply to a minimal model with only two sterile neutrinos, and therefore a single dim-five dipole coefficient $d$. We expect that, extending the analysis to the case of e.g. three sterile neutrinos, with three independent dipole coefficients and general active-sterile mixing, the sensitivities should qualitatively remain the same, as (i) we expect only an order one change in the total number of events and (ii) the upper bounds on $\theta_{i\alpha}$ roughly apply to each sterile state separately. On the other hand, the increased number of parameters would weaken the correlations that we illustrated, both between the dipole and the mixing, and among the different lepton flavours.

Finally, we remark that traditional constraints on sterile neutrinos, presented as sensitivities in the mass-mixing plane ($\theta_{i\alpha}$ versus $M_i$), might be modified by the presence of neutrino electromagnetic dipoles. In particular, we point out that the irreducible contribution $d_{N_i \nu_\alpha}^{EW}$, coming from EW loops, is sharply predicted, and it should be taken into account for a consistent analysis.

# Acknowledgments

We thank Riccardo Fantechi, Enrique Fernández Martínez, Daniel Naredo-Tuero for useful suggestions. We also thank Daniele Barducci, Marco Taoso, Christoph Ternes and Claudio Toni for the past collaborations that generated the meson spectra used in this work, and for reading the manuscript. MF wishes to thank the Instituto de Física da Universidade de São Paulo (IFUSP) for the warm hospitality during the initiation of this work.

**Funding information** MF received support from the European Union Horizon 2020 research and innovation program under the Marie Skłodowska-Curie grant agreements No 860881-HIDDeN and No 101086085-ASYMMETRY. The work of EB is partly supported by the Italian INFN program on Theoretical Astroparticle Physics (TAsP).

# A  Dipole-induced neutrino decay and Dirac limit

Here we present in some detail the derivation of Eq. (35), where the initial and final state neutrinos are Majorana fermions. We then clarify what happens when the initial state becomes a Dirac fermion (i.e. the limit when the mass splitting between $N_1$ and $N_2$ vanishes, $\delta \to 0$).

Consider two Weyl fermions, $\psi_1$ and $\psi_2$, with a dipole interaction

$$\mathcal{L}_{\text{dipole}} = d_\psi \psi_1 \sigma^{\mu\nu} \psi_2 F_{\mu\nu} + \text{h.c.} = \frac{1}{2}\overline{\Psi_1}\Sigma^{\mu\nu}\left(d_\psi P_L - d_\psi^* P_R\right)\Psi_2 F_{\mu\nu}, \qquad (A.1)$$

where in the last step we introduced four-component Majorana fermions and the associated antisymmetric tensor [45],

$$\Psi_i = \begin{pmatrix} \psi_i \\ \psi_i^\dagger \end{pmatrix}, \qquad \Sigma^{\mu\nu} = 2\begin{pmatrix} \sigma^{\mu\nu} & 0 \\ 0 & \bar{\sigma}^{\mu\nu} \end{pmatrix}.$$

From Eq. (A.1) we can immediately compute the amplitude for the $\psi_2 \to \psi_1 \gamma$ decay,

$$\mathcal{A} = \quad\quad\quad\quad\quad\quad = q_\mu \bar{u}_p \Sigma^{\mu\nu}(-\text{Re}\,d_\psi \gamma^5 + i\text{Im}\,d_\psi)u_Q \epsilon_\nu(q), \qquad (A.2)$$

and from this we obtain

$$\Gamma(\psi_2 \to \psi_1 \gamma) = \frac{|d_\psi|^2}{8\pi}m_2^3\left(1 - \frac{m_1^2}{m_2^2}\right)^3. \qquad (A.3)$$

In general, one can take into account possible loop corrections to the amplitude, that can be effectively included in $d_\psi = d_\psi^{tree} + d_\psi^{loop}$: in the case of neutrinos, EW loops do generate a relevant correction, studied in detail in appendix B. After the appropriate identification of $\psi_{1,2}$ with the neutrino mass eigenstates $N_{1,2}$ and $\nu_\alpha$, and of $d_\psi$ with the corresponding dipole coefficients, this result allows to reproduce the decay widths shown in Eq. (35).

Let us now focus on two sterile neutrinos where, with a little abuse of notation, we indicate with $N_{1,2}$ both the (left-handed) Weyl fermions and the corresponding (massive) Majorana fermions. In the limit $\delta \to 0$, $N_1$ and $N_2$ become mass degenerate, i.e. they form one Dirac fermion $N$ (with Weyl components $N_L = N_1$ and $N_R = N_2^\dagger$). The decay channel $N_2 \to N_1 \gamma$ is kinematically closed, and the remaining decay amplitudes rearrange in such a way that

$$\Gamma(N_1 \to \nu\gamma) = \Gamma(N_2 \to \nu\gamma) = \Gamma(N \to \nu\gamma) = \Gamma(\bar{N} \to \nu\gamma). \qquad (A.4)$$

Therefore, the sum of the widths of the two Majorana components is equal to the sum of the widths of the Dirac particle $N$ and antiparticle $\bar{N}$, as it must be in order for the number of produced photons to vary continuously, as one takes the limit $\delta \to 0$.

A similar reasoning applies to all other decay channels involving $N_{1,2}$, discussed in App. C. In the expressions for the decay widths, we generally take the convention where the Majorana particle $N_{1,2} \equiv \bar{N}_{1,2}$ includes both chiralities, and analogously $\nu \equiv \bar{\nu}$ stands for a light Majorana neutrino (including both the left-handed neutrino and the right-handed antineutrino).

# B  Neutrino dipole from electroweak loops

Even in the absence of a higher-dimension dipole operator, neutrinos do acquire a dipole coupling to the photon, via a loop involving a charged lepton and a $W$ boson (the photon can be

attached to either one or the other particle in the loop). Such EW effect induces an electro-magnetic dipole amplitude, $d_{jk}^{EW} = -d_{kj}^{EW}$, between any pair of neutrino mass eigenstates $\nu_j$ and $\nu_k$, as long as they have a non-zero mixing with the SM neutrinos $\nu_\alpha$ ($\alpha = e, \mu, \tau$). This allows for neutrino pair production from a photon, $\gamma^* \to \nu_j \nu_k$, as well as neutrino radiative decay, $\nu_k \to \nu_j \gamma$ (for $m_k > m_j$). In the presence of a higher-dimension NP operator, contribut-ing to the neutrino electromagnetic dipole with coefficient $d_{jk}^{NP}$, one must consider the sum of the two effects, $d^{em} \equiv d^{NP} + d^{EW}$. The radiative decay width $\Gamma(\nu_k \to \nu_j \gamma)$, in particular, is given by Eq. (A.3), with the obvious replacements $m_2 \to m_k$, $m_1 \to m_j$, and $d_\psi \to d_{jk}^{em}$.

Let us provide the general expression for $d^{EW}$. A neutrino flavour eigenstate can be written as $\nu_\alpha = \sum_{j=1}^n U_{\alpha j} \nu_j$, where the sum runs over all the neutrino mass eigenstates: $n = 3 + n_s$ with $n_s$ the number of sterile neutrinos (in the rest of this paper $n_s = 2$, with $\nu_{4,5} \equiv N_{1,2}$ and $m_{4,5} \equiv M_{1,2}$). The computation of the EW loops give [43, 44]

$$d_{jk}^{EW} = \frac{e G_F}{4\sqrt{2}\pi^2} \sum_{\alpha=e,\mu,\tau} f\left(\frac{m_\alpha^2}{m_W^2}\right) \left(m_k U_{\alpha j}^* U_{\alpha k} - m_j U_{\alpha j} U_{\alpha k}^*\right), \tag{B.1}$$

where

$$f(x) \equiv \frac{3}{4}\left[1 + \frac{1}{1-x} - \frac{2x}{(1-x)^2} - \frac{2x^2 \log x}{(1-x)^3}\right] = \frac{3}{2} - \frac{3}{4}x + \mathcal{O}(x^2). \tag{B.2}$$

Denoting the sterile neutrinos $\nu_{s_A}$ for $A = 1, \ldots, n_s$, the unitarity of the full neutrino mixing matrix implies

$$\sum_{\alpha=e,\mu,\tau} U_{\alpha j}^* U_{\alpha k} = \delta_{jk} - \sum_{A=1}^{n_s} U_{s_A j}^* U_{s_A k}. \tag{B.3}$$

In the absence of sterile neutrinos, the right-hand side is just the identity, so that the con-stant term $3/2$ in $f(x)$ gives a vanishing $d_{jk}^{EW}$. Therefore, the EW contribution to the dipole amplitude is suppressed by one power of $m_\alpha^2/m_W^2$.

In the presence of sterile neutrinos, instead, one avoids such chiral suppression. On the other hand, $d_{jk}^{EW}$ is suppressed by the small active-sterile mixing parameters, given by $U_{\alpha k}$ for $k > 3$. Using Eq. (B.3) one can check that, in the case of two light neutrinos ($j, k \le 3$), the dipole $d_{jk}^{EW}$ is suppressed by two powers of the active-sterile mixing. The same is true for two heavy neutrinos ($j, k > 3$). The most relevant case is the transition between one heavy $\nu_k$ and one light $\nu_j$, as $d_{jk}^{EW}$ turns out to be suppressed by only one power of the active-sterile mixing. In the limit of massless light neutrinos ($m_j = 0$ for $j = 1, 2, 3$), one can choose $U_{\alpha j} \simeq \delta_{\alpha j}$ and obtain

$$d_{jk}^{EW} \simeq \frac{3 e G_F}{8\sqrt{2}\pi^2} m_k U_{\alpha k} \delta_{\alpha j} \qquad (k > 3, \, j \le 3). \tag{B.4}$$

It is interesting to compare $d_{jk}^{EW}$ with the contribution coming from the dimension-five operator of Eq. (1). Moving to the basis of neutrino mass eigenstates and taking into account Eq. (4), one obtains

$$d_{jk}^{NP} = d \cos\theta_w (U_{s_1 j} U_{s_2 k} - U_{s_1 k} U_{s_2 j}). \tag{B.5}$$

In particular, choosing the explicit form of the mixing matrix $U$ given by Eqs. (24) and (31), the NP dipole between one heavy neutrino ($k = 4$ for $N_1$ and $k = 5$ for $N_2$) and one light neutrino ($j = 1, 2, 3$) is given by

$$d_{j4}^{NP} \simeq d \cos\theta_w \left(-\frac{i}{\sqrt{2}} s_\alpha \delta_{\alpha j} \sqrt{1-\delta}\right), \qquad d_{j5}^{NP} \simeq d \cos\theta_w \left(-\frac{1}{\sqrt{2}} s_\alpha \delta_{\alpha j} \sqrt{1+\delta}\right). \tag{B.6}$$

To compare with Eq. (B.4), notice that $U_{\alpha 4} \simeq -i s_\alpha \sqrt{1+\delta}/\sqrt{2}$ and $U_{\alpha 5} \simeq s_\alpha \sqrt{1-\delta}/\sqrt{2}$: the same, small mixing angle $\theta_\alpha$ enters in both the EW and NP components of $d_{jk}^{em}$.

## C  Expressions for the decay widths

Let us begin by introducing the various meson decay constants. For charged pseudoscalar ($P$) and vector ($V$) mesons, they are defined by

$$\langle 0|\bar{u}_i \gamma^\mu \gamma_5 d_j|P^-(p)\rangle = i f_P \, p^\mu \,, \qquad \langle 0|\bar{u}_i \gamma^\mu \gamma_5 d_j|V^-(p)\rangle = i f_V \, m_V \, \epsilon^\mu(p) \,, \qquad (C.1)$$

where $u_i$ and $d_j$ denote up and down-type quarks of flavour $i$ and $j$, while $m_V$ and $\epsilon^\mu(p)$ are the vector meson mass and polarisation vector. Numerical values for $f_{P,V}$ can be found in [53, 60].

For neutral pseudoscalar and vector mesons, what matters are the matrix elements of the $Z$ boson current, defined as

$$J_Z^\mu = \sum_f \left[ \bar{f} \gamma^\mu (T_L^3 - 2Q \sin^2 \theta_w) f - \bar{f} \gamma^\mu \gamma_5 T_L^3 f \right] \,, \qquad (C.2)$$

with $T_L^3$ the eigenvalue of the third $SU(2)_L$ generator, $Q$ the electric charge and $\theta_w$ the weak angle. The neutral decay constants are defined via

$$\langle 0|J_Z^\mu|P^0(p)\rangle = i \, \frac{f_P}{\sqrt{2}} p^\mu \,, \qquad \langle 0|J_Z^\mu|V^0(p)\rangle = i \, \frac{\kappa_V f_V m_V}{\sqrt{2}} \epsilon^\mu(p) \,. \qquad (C.3)$$

Numerical values of $f_P$ for the neutral pseudoscalar mesons, and of $f_V$ for the light vector mesons $\rho$, $\omega$ and $\phi$, can be found in Refs. [53, 60]. Notice that, in these cases, our constants $f_V$ are related to the constants $g_V$ defined in Ref. [53] by the expression $f_V = g_V/m_V$. For the quarkonia vector mesons $J/\psi$ and $\Upsilon$ (respectively, a $c\bar{c}$ and a $b\bar{b}$ bound state), it is instead convenient to express the decay constant $f_V$ in terms of the known decay width into electron-positron pairs. Using the results of [61], we have

$$\Gamma(V_{(q\bar{q})} \to e^+ e^-) = \frac{f_V^2 \, e^4 \, Q_q^2}{24\pi \, m_V} \,, \qquad (C.4)$$

where the notation $V_{(q\bar{q})}$ means that the vector meson is composed by a $q\bar{q}$ pair and $Q_q$ denotes the $q$ electric charge.

The factors $\kappa_V$ appearing in Eq. (C.3) take into account that the current associated with each vector-meson appears into the $Z$ current $J_Z^\mu$ with a specific coefficient. They are given by[5]

$$\begin{aligned}
\kappa_\rho &= 1 - 2\sin^2 \theta_w \,, \\
\kappa_\omega &= -\frac{2}{3} \sin^2 \theta_w \,, \\
\kappa_\phi &= -\sqrt{2}\left(\frac{1}{2} - \frac{2}{3}\sin^2 \theta_w\right) \,, \\
\kappa_{J/\psi} &= \frac{1}{2} - \frac{4}{3}\sin^2 \theta_w \,, \\
\kappa_\Upsilon &= -\frac{1}{2} + \frac{2}{3}\sin^2 \theta_w \,.
\end{aligned} \qquad (C.5)$$

---

[5]Our expressions for $\kappa_{\rho,\omega,\phi}$ agree with those found in [60] and not with those of [53]. The expression for $\kappa_{J/\psi}$ is not reported in [60] but it is shown in [53]. With respect to this reference, we find a value $\kappa_{J/\psi}$ a factor of 2 smaller. At last, $\kappa_\Upsilon$ is, to the best of our knowledge, considered here for the first time.

## C.1  Mixing-induced sterile neutrino decays

The sterile neutrino decay widths induced by the dipole operator were already presented in Eq. (35). Here we report explicit expressions for the mixing-induced decay widths of heavy sterile neutrinos into mesons and leptons. We combine the findings of Refs. [62], [53] and [60], verifying independently the results where differences appear.

Before showing the expressions for the sterile neutrino decay widths, we define the kinematical functions[6]

$$
\begin{aligned}
\lambda(a,b,c) &= a^2 + b^2 + c^2 - 2ab - 2ac - 2bc, \\
I_1(x,y) &= [(1+x-y)(1+x) - 4x]\sqrt{\lambda(1,x,y)}, \\
I_2(x,y) &= [(1+x-y)(1+x+2y) - 4x]\sqrt{\lambda(1,x,y)}, \\
J_1(x,y,z) &= 12\int_{(x+y)^2}^{(1-z)^2} \frac{ds}{s}(s - x^2 - y^2)(1 + z^2 - s)\sqrt{\lambda(s,x^2,y^2)}\sqrt{\lambda(1,s,z^2)}, \\
J_2(x,y,z) &= 24\,y\,z\int_{(y+z)^2}^{(1-x)^2} \frac{ds}{s}(1 + x^2 - s)\sqrt{\lambda(s,y^2,z^2)}\sqrt{\lambda(1,s,x^2)}.
\end{aligned}
\tag{C.6}
$$

For the two-body sterile neutrino decays we have

$$
\begin{aligned}
\Gamma(N_i \to \ell_\alpha^- P^+) &= \frac{G_F^2}{16\pi} f_P^2 \left|V_{q_u q_d}\right|^2 |\theta_{i\alpha}|^2 M_i^3\, I_1\!\left(\frac{m_{\ell_\alpha}^2}{M_i^2}, \frac{m_P^2}{M_i^2}\right), \\
\Gamma(N_i \to \nu_\alpha P^0) &= \frac{G_F^2}{16\pi} f_P^2 |\theta_{i\alpha}|^2 M_i^3\, I_1\!\left(0, \frac{m_P^2}{M_i^2}\right), \\
\Gamma(N_i \to \ell_\alpha^- V^+) &= \frac{G_F^2}{16\pi} f_V^2 \left|V_{q_u q_d}\right|^2 |\theta_{i\alpha}|^2 M_i^3\, I_2\!\left(\frac{m_{\ell_\alpha}^2}{M_i^2}, \frac{m_V^2}{M_i^2}\right), \\
\Gamma(N_i \to \nu_\alpha V^0) &= \frac{G_F^2}{16\pi} \kappa_V^2 f_V^2 |\theta_{i\alpha}|^2 M_i^3\, I_2\!\left(0, \frac{m_V^2}{M_i^2}\right),
\end{aligned}
\tag{C.7}
$$

where $G_F$ is the Fermi constant, while the subscripts in the CKM matrix element $V_{q_u q_d}$ denote the up-type quark $q_u$ and down-type antiquark $\bar{q}_d$ composing the corresponding charged meson. The decay widths in Eq. (C.7) will be used in the non-perturbative regime of QCD, which we will take to be $M_i \leq 1.6$ GeV.

In order to write an expression for the total decay width into one lepton plus hadrons, for masses $M_i > 1.6$ GeV, we follow Refs. [53,60] and write

$$
\Gamma(N_i \to \text{lepton+hadrons}) \simeq \left[1 + \Delta_{\text{QCD}}(M_i)\right]\Bigg[2\Gamma(N_i \to \ell_\alpha^- u\bar{d}) + 2\Gamma(N_i \to \ell_\alpha^- u\bar{s})
\tag{C.8}
$$

$$
+ \Gamma(N_i \to \nu_\alpha u\bar{u}) + \Gamma(N_i \to \nu_\alpha d\bar{d}) + \sqrt{1 - \frac{4m_{K^0}^2}{M_i^2}}\,\Gamma(N_i \to \nu_\alpha s\bar{s})\Bigg],
$$

where

$$
\Delta_{\text{QCD}}(x) = \frac{\alpha_s(x)}{\pi} + 5.2\frac{\alpha_s^2(x)}{\pi^2} + 26.4\frac{\alpha_s^3(x)}{\pi^3},
\tag{C.9}
$$

---

[6]To avoid confusion with the functions $I_{1,2}(x,y)$, we have renamed the functions called $I_{1,2}(x,y,z)$ in Ref. [62] as $J_{1,2}(x,y,z)$.

and the decay widths into quarks read

$$\Gamma(N_i \to \ell_\alpha^- q_u \bar{q}_d) = \frac{G_F^2}{192\pi^3} |V_{q_u q_d}|^2 |\theta_{i\alpha}|^2 M_i^5 J_1\left(\frac{m_{\ell_\alpha}}{M_i}, \frac{m_{q_u}}{M_i}, \frac{m_{q_d}}{M_i}\right), \tag{C.10}$$

$$\Gamma(N_i \to \nu_\alpha q\bar{q}) = \frac{G_F^2}{96\pi^3} |\theta_{i\alpha}|^2 M_i^5\left[\left(g_L^q g_R^q\right) J_2\left(0, \frac{m_q}{M_i}, \frac{m_q}{M_i}\right)\left((g_L^q)^2 + (g_R^q)^2\right) J_1\left(0, \frac{m_q}{M_i}, \frac{m_q}{M_i}\right)\right],$$

with

$$g_L^{q_u} = \frac{1}{2} - \frac{2}{3}\sin^2\theta_w, \qquad g_R^{q_u} = -\frac{2}{3}\sin^2\theta_w,$$
$$g_L^{q_d} = -\frac{1}{2} + \frac{1}{3}\sin^2\theta_w, \qquad g_R^{q_d} = \frac{1}{3}\sin^2\theta_w. \tag{C.11}$$

In Eq. (C.8), the factors of 2 take into account the decay into charge conjugate states, while the square root in the last term is introduced to effectively take into account the strange-mesons mass threshold. In Eq. (C.8) we also neglect heavy quark contributions, because they are phase space (and, in some cases, CKM) suppressed.

Turning to three-body decays into leptons, we have

$$\Gamma(N_i \to \ell_\alpha^- \ell_\beta^+ \nu_\beta) = \frac{G_F^2}{192\pi^3} |\theta_{i\alpha}|^2 M_i^5 J_1\left(0, \frac{m_{\ell_\alpha}}{M_i}, \frac{m_{\ell_\beta}}{M_i}\right),$$

$$\Gamma(N_i \to \nu_\alpha \ell_\beta^+ \ell_\beta^-) = \frac{G_F^2}{96\pi^3} |\theta_{i\alpha}|^2 M_i^5\left[\left(g_L^\ell g_R^\ell + \delta_{\alpha\beta} g_R^\ell\right) J_2\left(0, \frac{m_{\ell_\beta}}{M_i}, \frac{m_{\ell_\beta}}{M_i}\right)\right.$$
$$\left. \times \left((g_L^\ell)^2 + (g_R^\ell)^2 + \delta_{\alpha\beta}(1 + 2g_L^\ell)\right) J_1\left(0, \frac{m_{\ell_\beta}}{M_i}, \frac{m_{\ell_\beta}}{M_i}\right)\right], \tag{C.12}$$

$$\Gamma(N_i \to \nu_\alpha \nu\bar{\nu}) = \frac{G_F^2}{96\pi^3} |\theta_{i\alpha}|^2 M_i^5,$$

where, in the first decay rate, $\alpha \neq \beta$ (the case $\alpha = \beta$ is included in the second decay rate), while the couplings of the leptons to the $Z$ boson are given by

$$g_L^\ell = -\frac{1}{2} + \sin^2\theta_w, \qquad g_R^\ell = \sin^2\theta_w. \tag{C.13}$$

## C.2 Mesons decays

In order to compute the number of sterile neutrino events in the various experiments, it is necessary to know the decay widths of mesons into $N_{1,2}$. As already explained in Sec. 3, we consider only two-body decays, which are dominant.

The dipole-induced process $V^0 \to N_1 N_2$, with $V^0$ a vector meson, has decay width

$$\Gamma(V^0 \to N_1 N_2) = \frac{\left(|d| \cos\theta_w Q_q e f_V\right)^2}{24\pi} m_V \sqrt{\lambda\left(1, \frac{M_1^2}{m_V^2}, \frac{M_2^2}{m_V^2}\right)}$$
$$\times \left(1 + \frac{M_1^2 + M_2^2 + 6M_1 M_2 \cos(2\xi)}{m_V^2} - 2\frac{(M_2^2 - M_1^2)^2}{m_V^4}\right), \tag{C.14}$$

where $m_V$ is the vector meson mass, $Q_q$ is the charge of the constituent quarks, $e$ the electric charge, and we wrote the dipole coefficient as $d = |d|e^{i\xi}$. The $\xi$ dependence arises from the interference between the 'vector and axial' parts of the dipole coupling. Note that the decays $V^0 \to N_1 N_1$ and $V^0 \to N_2 N_2$ are forbidden because a Majorana-fermion vector current is anti-symmetric. The decays $P^0 \to N_i N_j$ are also forbidden, because the amplitude is proportional

to $q_{P0}^{\mu} q_{P0}^{\nu} \sigma_{\mu\nu} = 0$, where one momentum comes from the first matrix element in Eq. (C.3), and the other momentum comes from the derivative in the $Z$ field strength. Finally, we neglect the decays $V^0 \to \nu_\alpha N_i$ induced by the dipole, because they are additionally suppressed by the active-sterile mixing.

Coming to meson decays induced by the active-sterile mixing, the decay widths are

$$\Gamma(P^- \to \ell_\alpha^- N_i) = \frac{G_F^2 f_P^2 m_P^3 \left|V_{q_u q_d}\right|^2 |\theta_{i\alpha}|^2}{8\pi} \left[ \frac{M_i^2}{m_P^2} + \frac{m_\ell^2}{m_P^2} - \left( \frac{M_i^2}{m_P^2} - \frac{m_\ell^2}{m_P^2} \right)^2 \right] \sqrt{\lambda\left(1, \frac{M_i^2}{m_P^2}, \frac{m_\ell^2}{m_P^2}\right)},$$

$$\Gamma(P^0 \to \nu_\alpha N_i) = \frac{G_F^2 f_P^2 m_P^3 |\theta_{i\alpha}|^2}{32\pi} \frac{M_i^2}{m_P^2} \lambda\left(1, 0, \frac{M_i^2}{m_P^2}\right), \tag{C.15}$$

$$\Gamma(V^0 \to \nu_\alpha N_i) = \frac{G_F^2 \kappa_V^2 f_V^2 m_V^3 |\theta_{i\alpha}|^2}{96\pi} \left(2 - \frac{M_i^2}{m_V^2} - \frac{M_i^4}{m_V^4}\right) \sqrt{\lambda\left(1, 0, \frac{M_i^2}{m_V^2}\right)},$$

where the decay widths of the neutral mesons are defined to include the contributions from both the SM neutrino and antineutrino. The charged pseudoscalar decay width has been also presented in [53, 60], and we agree with the expression reported in those papers.

## C.3  Comparison between dipole- and mixing-induced processes

We compare in Fig. 6 the relative contribution of the dipole and of the active-sterile mixing to the production and decay of sterile neutrinos, as a function of the sterile mass. Concerning production, we focus on the SHiP experiment for definiteness. This comparison is useful to understand the behaviour of the blue and orange lines shown in Figs. 1–2. In Fig. 6 we take a sterile mass splitting $\delta = 0.1$ and a dipole coefficient $d = 2.3 \times 10^{-6}$ GeV$^{-1}$, that corresponds

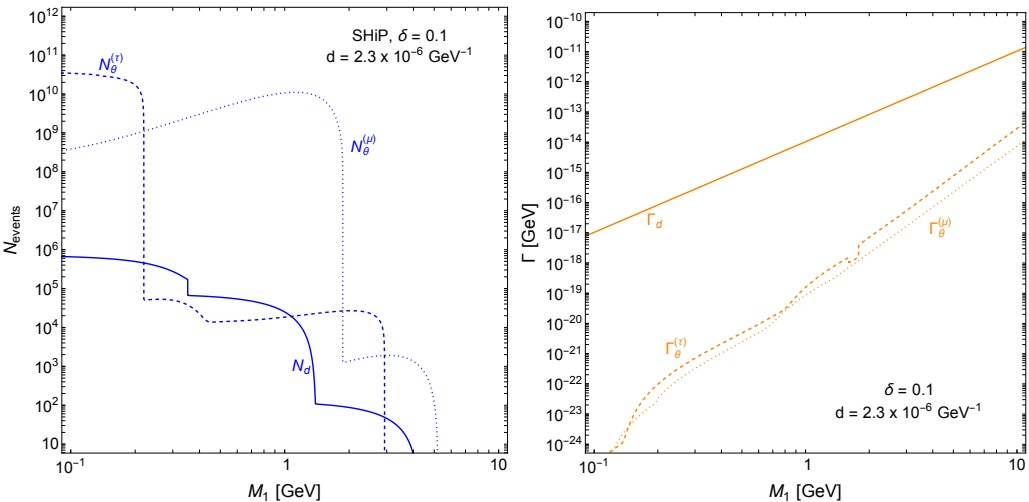

Figure 6: Left panel: number of $N_2$ particles produced at the SHiP experiment generated by the dipole, for $d = 2.3 \cdot 10^{-6}$ GeV, and by the mixing, for the maximal allowed value of $\theta_{i\alpha}$ (with two choices of flavour, see main text). The dipole (mixing) curve scales as $|d|^2$ ($|\theta_{2\alpha}|^2$). Right panel: decay width of the $N_2$ particle generated by the dipole and by the mixing (again, with two choices of flavour). The mixing curve scales as $|\theta_{2\alpha}|^2$, while the behaviour of the dipole curve is more involved, given the contribution of terms that scale as $|d|^2$ ($N_2 \to N_1 \gamma$), $|d|^2 |\theta_{2\alpha}|^2$ (NP part of $N_2 \to \nu_\alpha \gamma$) and $|\theta_{2\alpha}|^2$ (EW part of $N_2 \to \nu_\alpha \gamma$).

to the parameterisation of Eq. (2) with the choice $g_\star = 1$ and $m_\star = 1$ TeV. Moreover, we consider a mixing with either the $\mu$ or the $\tau$ flavours, fixing the mixing angles to their maximum allowed value at the reference point $M_1 = 1$ GeV, as in Sec. 3: $\theta_{1\mu} = 3 \times 10^{-4}$ and $\theta_{1\tau} = 10^{-3}$, respectively. We again focus on the production and decay of the sterile neutrino $N_2$.

The left panel shows the total number of events generated by the dipole (continuous line) and the mixing (dashed line for mixing with the $\tau$ flavour, dotted line for the mixing with the $\mu$ flavour) at the SHiP experiment, defined as

$$N_x = \sum_j N_{\mathbf{M}_j} \mathrm{BR}_x(\mathbf{M}_j \to N_2 + \dots). \tag{C.16}$$

In the expression above, $N_{\mathbf{M}_j}$ is the number of mesons produced at the SHiP experiment (see Eq. (D.3)), while $\mathrm{BR}_x(\mathbf{M}_j \to N_2 + \dots)$ denotes, schematically, the branching ratio of the meson $\mathbf{M}_j$ into $N_2$, for $x$ equal to $\theta$ (*dipole*) or $m$ (*mixing*). More concretely, when $x = \theta$ we consider the processes in Eq. (C.14), while for $x = m$ we consider the processes listed in Eq. (C.15). As can be seen, the number of events produced in the two cases have very different dependence on the sterile mass. This can be easily understood remembering that different mesons contribute in the two cases, with thresholds that appear for different values of $M_1$. We also observe that the behaviour of $N_\theta$ depends strongly on the flavour to which the sterile neutrino couples to. We remark that, in order to compute $N_d$ and $N_\theta$ for other values of the dipole and mixing parameters, it is sufficient to rescale $N_d$ by $|d|^2$[7] and $N_\theta$ by $|\theta_{i\alpha}|^2$. This translates into the behaviour of the blue curves in Figs. 1-2.

Turning to the right panel of Fig. 6, we show the dipole and mixing-induced decays widths, where $\Gamma_d$ is given by the sum of the first two lines of Eq. (35), while $\Gamma_\theta$ is given by the sum of the decay widths listed in App. C.1. The dipole-induced decay width $\Gamma_d$ scales as $M_1^3$. The same happens for $\Gamma_\theta$ at low masses, where the two-body decays of Eq. (C.7) dominate.[8] On the contrary, for $M_1 \gtrsim 2$ GeV, the mixing-induced decay widths scale with $M_1^5$, which is the dependence observed in the three-body decays of Eqs. (C.8) and (C.12). For our choice of parameters, the dipole always dominates over the mixing, but of course this will change if smaller values of the dipole coefficient are considered. The curves for $\Gamma_\theta$ for different values of $\theta_{i\alpha}$ can be obtained by simply rescaling with $|\theta_{i\alpha}|^2$. The scaling of $\Gamma_d$ is more involved, since there are terms that scale as $|d|^2$ (due to the $N_2 \to N_1 \gamma$ contribution), $|d|^2 |\theta_{2\alpha}|^2$ (NP contribution to $N_2 \to \nu_\alpha \gamma$) and $|\theta_{2\alpha}|^2$ (EW contribution to $N_2 \to \nu_\alpha \gamma$), so that an explicit computation is required. This translates into the behaviour of the orange curves in Figs. 1-2.

## D   Experiments geometry and analysis

We describe here the characteristics of the experiments considered that are important for our simulations. We start from the past experiments and then turn to the future proposals:

- **CHARM-II** was a proton beam dump experiment in which a 450 GeV proton beam was dumped on a beryllium target. The calorimeter that allowed for photon identification was placed at a distance of 780 m from the Interaction Point (IP) and had a transverse area of $(3.7 \times 3.7)\,\mathrm{m}^2$ and a length of 35.6 m [7]. In our simulation, we require photon production inside the calorimeter. During its run, CHARM-II collected a total number of Protons On Target (POT) equal to $N_{\mathrm{POT}} = 4.5 \times 10^{19}$.

---

[7]This is true as long as $d$ is sufficiently small that the total decay width of the meson is dominated by the SM contribution. If this is the case, the branching ratio in Eq. (C.16) inherits the $|d|^2$ from the decay width that appears in the numerator.

[8]This is true in the limit of massless final-state particles, but in practice the thresholds can be important, see right panel of Fig. 6.

- **BEBC** was a bubble chamber experiment in which a 400 GeV proton beam was dumped on a copper target [6]. The detector was placed at 404 m from the IP, with transverse area $(3.57 \times 2.52)\,\text{m}^2$ and a length of 1.85 m. The experiment collected $N_{\text{POT}} = 2.72 \times 10^{18}$.

- **NA62** is a currently running experiment at CERN, devised to study kaon physics. What concerns us is the planned beam-dump-mode, in which a 400 GeV proton beam will be dumped on a copper target. We take the geometry from [9] and assume that the decay volume will be a 1 m radius cylinder, aligned with the beam axis, placed at 79 m from the IP. At a distance of 138 m from the beginning of the decay volume is to be placed a cylindrical electromagnetic calorimeter of radius 1.2 m. In our simulation, we require the photons to be produced inside the decay volume with a trajectory that has a maximum angle $\theta_{\text{max}} = 5.5 \times 10^{-3}$ with respect to the beam axis. This guarantees that the photons produced will intersect the calorimeter, even when they are produced at the beginning of the decay volume, thus leading to a very conservative estimate of the sensitivity. We consider a dataset of $N_{\text{POT}} = 10^{18}$.

- **SHiP** will be a beam dump placed at CERN, in which a 400 GeV proton beam will collide with a molybdenum/tungsten target. Following [8], the SHiP decay volume will have a squared-base pyramid shape, that for simplicity we approximate to a truncated cone of angle $\theta_{\text{max}} = 2.8 \times 10^{-2}$. The vertex of the cone is taken at the IP, while the decay volume starts at a distance of 33 m and has a length of 50 m. We assume that the total number of POT collected at the end of the run will be $N_{\text{POT}} = 6 \times 10^{20}$.

- **FASER 2** is the proposed upgrade of the currently running FASER. If approved, it will be placed in a cavern near the ATLAS IP at the LHC, surrounded by rock and dirt that will shield the detector. For the geometry, we follow [10] and take the decay volume to be a cylinder of 1 m radius and 10 m length, to be placed at 620 m from the IP. The decay volume will be followed by a 10 m long tracking system, after which a calorimeter will be placed. In our analysis, we consider photons produced from the beginning of the decay volume up to the beginning of the calorimeter, and require the maximum angle between the photon momentum and the experiment axis to be $\theta_{\text{max}} = 1.6 \times 10^{-3}$ in order for the calorimeter to be intercepted. FASER 2 is expected to collect a total luminosity of 3 $\text{ab}^{-1}$.

The total number of mesons of type $M$, $\mathcal{N}_M$, that enters in Eqs. (37)–(44), can be computed in terms of the multiplicity of mesons produced per proton interaction, $f_M$, as

$$\mathcal{N}_M \equiv N_{\text{POT}} f_M \quad \text{(beam dump experiments)}, \tag{D.1}$$

or

$$\mathcal{N}_M \equiv \sigma_{\text{inelastic}} \mathcal{L} f_M \quad \text{(FASER 2)}, \tag{D.2}$$

where $\sigma_{\text{inelastic}} = 79.5$ mb is the inelastic cross section for proton-proton collisions at a center-of-mass energy $\sqrt{s} = 13$ TeV and $\mathcal{L} = 3\,\text{ab}^{-1}$ is the expected total luminosity. For the beam dump experiments, the mesons multiplicity and spectra have been simulated using PYTHIA 8 [63,64], as described in [22]. We obtain

$$\begin{aligned}
f_{\pi^0} &= 4.3\,, & f_\eta &= 0.49\,, & f_{\eta'} &= 0.055\,, \\
f_\rho &= 0.58\,, & f_\omega &= 0.57\,, & f_\phi &= 0.021\,, \\
f_{D^\pm} &= 4.3 \times 10^{-4}\,, & f_{D_s} &= 1.8 \times 10^{-4}\,, & f_{B^\pm} &= 6.0 \times 10^{-8}\,, \\
f_{J/\psi} &= 4.7 \times 10^{-6}\,, & f_\Upsilon &= 2.2 \times 10^{-9}\,.
\end{aligned} \tag{D.3}$$

We neglect the small differences between the 400 GeV proton beam (CHARM-II, SHiP, NA62) and the 450 GeV proton beam (BEBC). We checked that they are indeed small. For FASER 2, we instead take the mesons spectra and multiplicity from the FORESEE package [65].

Meson spectra are important because they allow to produce spectra for $N_{1,2}$ and the photons produced in the decay. In practice, we proceed as follows: we simulate $N_{1,2}$ 4-momenta in the meson center-of-mass frame and then use the information about the meson spectra to boost these 4-momenta in the laboratory frame. We apply the same procedure also to simulate the photon spectra: we first create a list of photon 4-momenta in the $N_{1,2}$ center-of-mass frame, and then use the $N_{1,2}$ spectra to boost to the laboratory frame. The photon spectra are used to compute the probability $P_{E_\gamma}^X$ introduced in Sec. 3.2.

Finally, let us comment on the photon threshold required to produce a signal. For the past beam dump experiments CHARM-II and BEBC, we take $E_\gamma \in [3, 24]$GeV [66] and $E_\gamma > 1$ GeV [6], respectively. For the future experiments, we follow Refs. [59, 67, 68] and require $E_\gamma > 1$ GeV.

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
