# Peer review of "Two portals to GeV sterile neutrinos : dipole versus mixing"

_SciPost Physics, doi:SciPost Phys. 18, 140 (2025)_

## Round 1 · Referee Report · Anonymous (Referee 1) · 2025-3-5

Report

I have reviewed the manuscript entitled "Two portals to GeV sterile neutrinos: dipole versus mixing," submitted for publication in SciPost Physics.

The authors provide a comprehensive analysis of the interplay between two different channels for coupling SM singlet fermions to the SM degrees of freedom. One is via active--sterile mixing (or the so-called neutrino portal), the other is through a higher-dimensional magnetic-moment operator involving active and sterile neutrinos. They concentrate on the mass window defined by 100 MeV and 10 GeV, and concentrate their phenomenological discussions to beam-dump-like facilities, including SHiP and FASER2 (FASER2 is not a really a beam dump facility, of course).

The manuscript is well organized and the presentation is clear. While the simulations are somewhat simplified, the results are sound and useful; I am happy to recommend the manuscript for publication. I do have a few comments and questions that the authors should consider, however. These are not ranked in any particular way. Regarding the journal's acceptance criteria, as flagged by the authors upon submission, this manuscripts explores the interplay of two channels connecting the SM to new physics (the sterile states) and discusses how different types of experiments complement one another so, in some sense, they "Provide a novel and synergetic link between different research areas." There is also opportunity for follow-up research and generalizations of the scenarios discussed here.

  • Equation (2.1) seems a little redundant. I think you can define $N_1'$ and $N_2'$ in such a way that M is zero. I am not this is useful, however, but I will leave that to the authors to decide.

  • Below Equation 2.5, I presume $A = 1,2, 3$ and that $(m_Am_A)^{1/2}$ has an implied sum over $A$?

  • Throughout the model discussion, I assume the authors have assumed that contributions from the Weinberg operator are negligible. Since they include the dipole operators and the same physics that contributes to the dipole operators can also contribute to the Weinberg operator, they may want to comment on the self-consistency of this choice. I don't think there are any problems here, but it is useful to discuss it.

  • Are there any constraints on the invisible dipole from the $Z$? For example, $Z\to N_1 N_2$? I suspect the answer is no, but I am not at all sure. I don't recall an analysis along these lines and the effect could be very tiny...

Recommendation

Publish (meets expectations and criteria for this Journal)

  • validity: -
  • significance: -
  • originality: -
  • clarity: -
  • formatting: -
  • grammar: -

Author:  Michele Frigerio  on 2025-03-13  [id 5289]

(in reply to Report 1 on 2025-03-05)

We provide our reply in the pdf attached.

Attachment:

Reponse_sterile_dipole.pdf

---

## Round 1 · Referee Report · Anonymous (Referee 2) · 2025-3-21

Report

Referee report for "Two portals to GeV sterile neutrinos: dipole versus mixing" submitted for publication in SciPost Physics.

The authors study the interplay between two portals related to heavy neutral leptons (HNL): their mixing with active neutrinos, and a dipole coupling to photons. They proceed to discuss how the sterile neutrinos, with masses from 0.1 GeV to 10 GeV, can be probed at experiments such as NA62, SHiP, and FASER2. In particular, the sensitivity of the SHiP experiment to very low dipole couplings is highlighted.

The authors analyze HNL production in meson decays, and their subsequent decays into photons. The manuscript fills a gap in the existing literature by accounting for the correlations between the dipole and mixing. The analysis appears technically sound, and the presented results are relevant for future experiments.

After the questions raised by referee 1, and the authors reply to them (which I am happy with), I only have a couple of additional minor suggestions for the authors to consider:

  • In Figs. 1 and 2 the transparent yellow and blue form a reddish shade when overlapping, which is initially confusing as the shade is undocumented in the legend. If the plots were to be shown in isolation, e.g. in a talk, some of them would not show any blue, but only yellow and red. I suggest changing either one of the colors to something more neutral, such as transparent gray or a hashed or dotted design, in order to enhance readability.

  • On page 16 the authors state that "Details on the experiments geometry, meson multiplicity and analysis can be found in Sec. 4 of Ref. [22]". The design of future experiments's (such as SHiP) has evolved over time since the first time they were proposed. Likewise, currently running experiments may indicate preference for certain predictions for meson production in the relevant kinematic regime over some alternatives used in the past. Since this is not a letter, where length would be a concern, I believe it would be the most transparent practice to list the most recent experiment-specific references briefly in each manuscript, instead of referring to another article with the details.

With these, I believe this the quality of the manuscript would improve, and I recommend the paper for publication.

Recommendation

Publish (meets expectations and criteria for this Journal)

  • validity: -
  • significance: -
  • originality: -
  • clarity: good
  • formatting: excellent
  • grammar: reasonable

Author:  Michele Frigerio  on 2025-03-27  [id 5322]

(in reply to Report 2 on 2025-03-21)
Category:
answer to question

Dear Editor and Referee,

we would like to thank the Referee for their suggestions:

  • In Figs. 1 and 2 the transparent yellow and blue form a reddish shade when overlapping, which is initially confusing as the shade is undocumented in the legend. If the plots were to be shown in isolation, e.g. in a talk, some of them would not show any blue, but only yellow and red. I suggest changing either one of the colors to something more neutral, such as transparent gray or a hashed or dotted design, in order to enhance readability.

We decided to keep the color code as it is but to enlarge the legend to include all possible cases (blue, orange, wine and white regions) to increase clarity. We have also updated the notation to make it more compact: instead of the subscript "dipole" we now use simply "d", while instead of "mixing" we now use "\theta".

  • On page 16 the authors state that "Details on the experiments geometry, meson multiplicity and analysis can be found in Sec. 4 of Ref. [22]". The design of future experiments (such as SHiP) has evolved over time since the first time they were proposed. Likewise, currently running experiments may indicate preference for certain predictions for meson production in the relevant kinematic regime over some alternatives used in the past. Since this is not a letter, where length would be a concern, I believe it would be the most transparent practice to list the most recent experiment-specific references briefly in each manuscript, instead of referring to another article with the details.

To answer this point, we have added a new appendix D in which we list the details needed to reproduce our simulations: updated experiment geometries with corresponding references (that can also we found in the text where the experiments are first cited), details on the computation of the meson spectra and multiplicities, discussion of the photon threshold.

We hope that, with these modifications, the paper will be suitable for publication in SciPost.

Faithfully, the Authors

---

## Round 3 · List of Changes

• the contribution of electroweak loops to the neutrino dipole is added, and it turns out to be within the reach of SHiP
  • details are added on the experiments geometry and analysis

---

## Editorial Decision

published